# CityBench: Evaluating the Capabilities of Large Language Models for Urban Tasks

## Abstract

Recently, large language models (LLMs) with extensive general knowledge and powerful reasoning abilities have seen rapid development and widespread application. A systematic and reliable evaluation of LLMs or visual language model (VLMs) is a crucial step in applying and developing them for various fields. There have been some early explorations about the usability of LLMs for limited urban tasks, but a systematic and scalable evaluation benchmark is still lacking. The challenge in constructing a systematic evaluation benchmark for urban research lies in the diversity of urban data, the complexity of application scenarios and the highly dynamic nature of the urban environment. In this paper, we design *CityBench*, an interactive simulator based evaluation platform, as the first systematic benchmark for evaluating the capabilities of LLMs for diverse tasks in urban research. First, we build *CityData* to integrate the diverse urban data and *CitySimu* to simulate fine-grained urban dynamics. Based on *CityData* and *CitySimu*, we design 8 representative urban tasks in 2 categories of perception-understanding and decision-making as the *CityBench*. With extensive results from 30 well-known LLMs and VLMs in 13 cities around the world, we find that advanced LLMs and VLMs can achieve competitive performance in diverse urban tasks requiring commonsense and semantic understanding abilities, e.g., understanding the human dynamics and semantic inference of urban images. Meanwhile, they fail to solve the challenging urban tasks requiring professional knowledge and high-level reasoning abilities, e.g., geospatial prediction and traffic control task. These observations provide valuable perspectives for utilizing and developing LLMs in the future. The dataset, benchmark and source codes are openly accessible to the research community via https://github.com/CityBench24/CityBench.

## 1 Introduction

Recent years, large language models (LLMs) with extensive commonsense and reasoning capabilities have achieved excellent results in various fields Achiam et al. (2023); Touvron et al. (2023), including programming Hong et al. (2023), mathematics Wei et al. (2022), visual intelligence Liu et al. (2024c) and commonsense reasoning Suzgun et al. (2022); Mialon et al. (2023). Furthermore, powerful LLMs enable many unimaginable research endeavors to become feasible, e.g., agent Wang et al. (2024a), embodied intelligence Reed et al. (2022) and AGI Feng et al. (2024b). These researchers postulate that LLMs, by acquiring extensive world knowledge and commonsense, hold the key to unlocking promising outcomes in these challenging applications. Many works Achiam et al. (2023); Gurnee & Tegmark (2023); Roberts et al. (2023) have demonstrated that LLMs can be regarded as 'world models' of our life and they are skilled at solving a wide variety of tasks across multiple fields, while other works Xiang et al. (2024); Yang et al. (2023); Wang et al. (2024b) indicate that LLMs lack an comprehensive understanding of the real physical world and fail to handle many real-life problems. However, these research efforts have primarily focused on the indoor environment Puig et al. (2018), while neglecting the outdoor environment, specifically the broader urban environment Batty et al. (2012); Zheng et al. (2014).

Various works have explored the potential of LLMs in modeling urban space and solving urban tasks. For example, researchers evaluate the potential of LLMs on remote sensing understanding tasks Kuckreja et al. (2023); Zhang & Wang (2024) and urban visual tasks Yan et al. (2024); Hao et al. (2024). Gurnee et al. Gurnee & Tegmark (2023) and Roberts et al. Roberts et al. (2023) evaluate

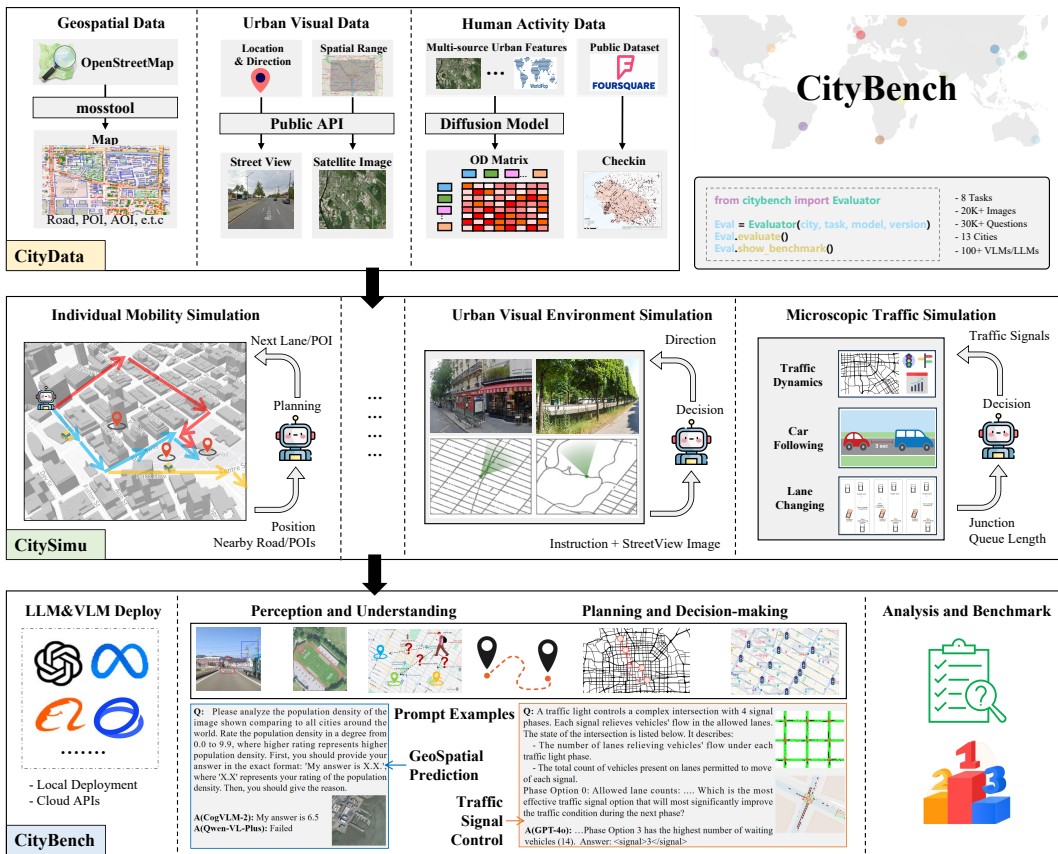

Figure 1: The framework of **CityBench**, which consists of a data collector *CityData*, an activity simulator *CitySimu* and 8 diverse urban tasks with different modalities. The evaluation data in the benchmark is collected from 13 cities around the world.

whether LLMs acquire the spatial knowledge of the world, such as cities and coordinates. Manvi et al. Manvi et al. (2023; 2024) try to extract the geospatial knowledge in LLMs to conduct geospatial indicator prediction tasks Mai et al. (2023). Besides, researchers also explore how to apply LLMs into the realistic urban applications, e.g., traffic control Da et al. (2023); Lai et al. (2023), traffic prediction Liu et al. (2024a), mobility prediction Wang et al. (2023b), visual language navigation Schumann et al. (2024) and so on. However, on the one hand, these existing works primarily focus on evaluating the static spatial knowledge of LLMs without considering the environment dynamics and interactivity. On the other hand, most of them only focus on one type of task and one modality of data in the urban space, using small dataset that are not scalable globally. Although there are some existing simulators for urban space such as game simulators Interactive (2023) and traffic simulators Lopez et al. (2018), they cannot be directly applied to support the evaluation and significant amount of adaptation work is required. None of them can support the systematic evaluation of LLMs' capabilities for diverse tasks in urban research, ranging from understanding and reasoning to decision-making tasks.

In this paper, we propose **CityBench**, a comprehensive evaluation platform for assessing the capabilities of LLMs to solve the diverse urban tasks. It covers multiple modalities, supports interactive simulations, and includes data from 13 cities around the world. *CityBench* consists of three modules: a data module *CityData* for collecting and processing diverse urban data, a simulation module *CitySimu* for simulating fine-grained urban dynamics, a evaluation module *CityBench* for the final evaluation of LLMs and VLMs. In *CityData*, we first collect three kinds of open urban data: geospatial data from Open Street Map, urban visual data from the Google map and ArcGIS, and human activity data from Foursquare and other websites. Then, we build an efficient simulation engine *CitySimu* to simulate fine-grained urban dynamics and develop various interfaces for controlling the urban dynamics and sensing the urban environments. Furthermore, based on *CitySimu*, we design a comprehensive benchmark to evaluate the capability of LLMs and VLMs, covering core research problems from various urban research fields. The benchmark comprises two levels of tasks:

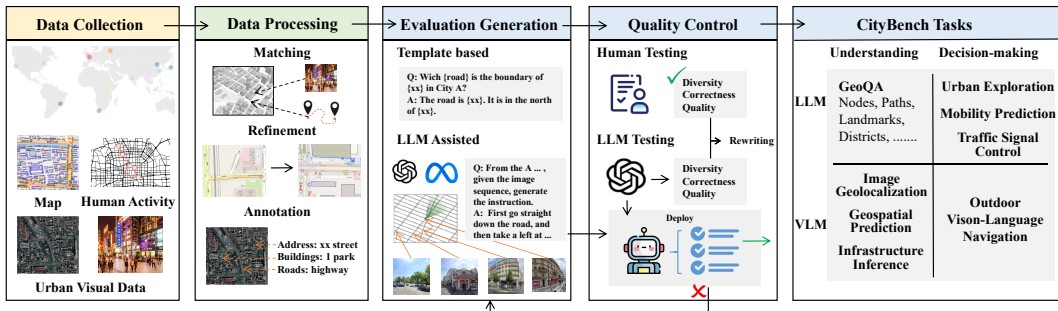

Figure 2: The pipeline of building benchmark, including data collection stage, data integration stage, evaluation generation stage and quality control stage.

perception&understanding tasks and decision-making tasks. In perception&understanding tasks, based on the integrated multi-source data from *CitySimu*, we introduce street view and satellite image understanding and urban space understanding tasks to evaluate the urban geospatial knowledge of LLMs and VLMs. In decision-making tasks, we apply LLMs and VLMs to interact with *CitySimu* to complete the urban exploration, visual navigation, mobility prediction and traffic signal control task, which require the comprehensive ability of them. In summary, our contribution are as follows,

- We develop *CityData* and *CitySimu*, an urban data collector and processor designed to support diverse urban tasks and applications, as well as an efficient urban simulator for generating find-grained urban dynamics. They provide ease-to-use APIs for controlling urban dynamics and sensing urban environments.
- We propose *CityBench*, a comprehensive evaluation benchmark for evaluating the capability of LLMs and VLMs for urban tasks, which includes 4 geospatial understanding tasks and 4 interactive urban decision-making tasks in 13 cities around the world.
- Extensive experiments on ***CityBench*** with 30 well-known open source and commercial LLMs and VLMs demonstrate the effectiveness of *CityBench* as evaluation benchmark and also discuss the potential and limitation of applying LLMs and VLMs in urban tasks, ranging from understanding and reasoning to decision-making task. Our results shed light for the further application and development of LLMs and VLMs in the future.

## 2 RELATED WORK

**LLM Evaluation for Urban Research** Researchers from various urban related fields have conducted extensive evaluations of LLM in urban space from different aspects. Zhang et al Zhang & Wang (2024) and Kuckreja et al. Kuckreja et al. (2023) evaluate the performance of multi-modal LLMs on several remote sensing related tasks. Yang et al. Yang et al. (2024) propose V-IRL benchmark to evaluate the performance of multi-modal LLMs on street view image related tasks including localization and recognition tasks. Mai et al. Mai et al. (2023) and Manvi et al. Manvi et al. (2023) use LLMs to predict social indicators like population and education level. Gurnee et al. Gurnee & Tegmark (2023) and Bhandari et al. Bhandari et al. (2023) try to testify whether LLMs know the coordinates of geospatial entity. Mooney et al. Mooney et al. (2023) and Deng et al. Deng et al. (2023) use GIS exams to understand the geospatial skills of LLMs. Roberts et al. Roberts et al. (2023) design GPT4GEO to evaluate the geospatial capabilities of LLM with limited case studies. Different from these works, we first introduce the interactive simulator based systematic evaluation system for LLMs and VLMs, which covers various data modalities, diverse urban task types and differentiated data from 13 cities around the world.

**Interactive Decision-making and Urban Simulator** Beyond the above static evaluation, researchers also evaluate the capacity of LLMs in the interactive decision making tasks with customized simulators, e.g., web agent Liu et al. (2023); Zeng et al. (2023) with web environment and embodied intelligence Yang et al. (2023) with virtual home Puig et al. (2018). In the urban domain, Schumann et al. Schumann et al. (2024) apply LLM to do the visual language navigation task in Touchdown Chen et al. (2019) and Lai et al. Lai et al. (2023) apply LLMs as the traffic light controller in CityFlow Zhang et al. (2019) to manage the road traffic. Besides, Yang et al. Yang et al. (2024) design V-IRL as the environment of street view image related tasks and propose a global scale virtual intelligence benchmark. These works only evaluate the potential of LLMs in single urban decision-making task

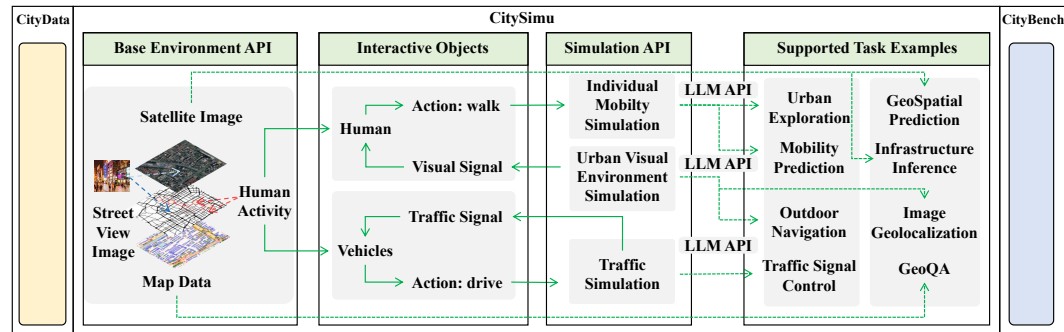

Figure 3: The simulation framework of *CitySimu*, including base environment APIs, interactive objects, simulation APIs and language APIs. Besides, supported task examples also present the relation between simulation APIs and evaluation tasks.

and most of their results rely on small-scale datasets in limited regions. Different from them, our work builds on an efficient urban simulator with global scale and supports 4 representative urban decision-making tasks with different modality in one benchmark, including urban exploration, visual navigation, mobility prediction and traffic control task.

## 3 METHODS

As presented in Figure 1, **CityBench** is a simulator based evaluation platform with three core components: *CityData* for collecting and processing diverse urban data, *CitySimu* for simulating human dynamics and providing an interactive simulation environment, and *CityBench* for model evaluation on 8 representative urban tasks with different modalities.

### 3.1 CITYDATA

To present a complete picture of the city's geospatial structure, semantic features, and human activities, *CityData* integrates the following globally available data from multiple sources.

**Geospatial Data** Geospatial data, represented by maps, is the most fundamental data for describing the urban structure including road networks, points of interest (POIs), areas of interest (AOIs), etc. OpenStreetMap (OSM) [1] is most widely used open source map data. However, the raw data provided by OSM cannot support the simulation of urban dynamics directly because the relationship between different elements is incomplete such as the connection between buildings and roads. Therefore, we provide a globally available rule-based map building tool within *CityData* that reconstructs lanes, lane topology, and building-lane connections based on the raw OSM data. The reconstructed map is used as the geospatial base and simulation input in *CitySimu*.

**Urban Visual Data** Street view data and satellite images are two types of globally available urban data that contains rich semantic information, which represents the visual of human. Therefore, *CityData* also integrates the two types of data, the former obtained via Google Maps API and Baidu Maps API, and the latter using the Esri World Imagery as data source. In the open-source *CityData* python package, street view data is accessed through spatial location and facing direction, and satellite images are acquired through spatial ranges.

**Human Activities Data** We use the global Foursquare-checkin Yang et al. (2016) data and a synthetic global origin-destination data (OD data) loooffeeeey (2024) as the proxy of human activities to enable the fine-grained human movement simulation. The Foursquare-checkin dataset Yang et al. (2016) is a long-term user check-in dataset collected from Foursquare [2] in approximately 400 cities worldwide. It has been widely used in the community over the past ten years Chen et al. (2024). Origin-destination data is generated by a diffusion model with population from Worldpop[3] and satellite image from Esri World Imagery as input. While all the user information are anonymized, we follow the license from Foursquare-checkin Yang et al. (2016) to protect the public privacy.

---

[1] https://openstreetmap.org/

[2] https://foursquare.com/

[3] https://www.worldpop.org/

## 3.2 CITYSIMU

Building on *CityData*, *CitySimu* simulates the urban dynamics and provide diverse easy-to-use APIs for the interactive operation. As shown in Figure 3, *CitySimu* contains the base environment APIs for obtaining the static information of environment, three simulation APIs for human and vehicle behavior simulations, language APIs to enable the interaction between *CitySimu* and LLMs.

**Individual Mobility Simulation** Based on the geospatial data, the individual mobility simulation constructs a simulator that can simulate an agent moving and exploring within the city. Agents can obtain the POIs and roads around them through API provided by *CityData*, and thus plan and decide the next lane or POI to travel in to update their locations. For the mobility prediction task in the city scale, the available actions are defined as the POIs around the city. For the urban exploration task in the local street scale, the available actions are defined as the nearby lanes.

**Urban Visual Environment Simulation** To further support the study of urban visual intelligence Fan et al. (2023), we follow Chen et al. (2019); Mirowski et al. (2018) to construct a urban visual environment simulation with real street view images and map data. In the environment, agent can access the panoramic images of its location via APIs and then select the available actions to move along the road to arrive the destination. In the outdoor visual-language instruction navigation task, given the human-like instruction, agent can observe the panoramic images of its location, extract key elements from them and then decide one direction to go. This can be saw as an extension of individual mobility simulation with visual input.

**Traffic Simulation** In the former two simulations, we only simulate the individual actions without the interaction with others. Here, we introduce microscopic traffic simulation to model the interaction behaviors between vehicles and provide a traffic control environment. The simulator takes the geospatial data reconstructed from OSM within *CityData* and the travel demand described by the synthetic global OD data as inputs. It simulates the vehicle behaviors through realistic driving simulation models including the intelligent driver model (IDM) Treiber et al. (2000) as the car-following model and the randomized MOBIL model Kesting et al. (2007); Feng et al. (2021) as the lane-change model to obtain the dynamics of all vehicles in the city at each second. Through the sensing APIs, LLMs can obtain data about urban dynamics such as junction queue length, vehicle speed, and road average speed. Through the control APIs, LLMs can intervene in the city's operation, such as modifying traffic signal lights, modifying the speed limit of the road, etc.

## 3.3 CITYBENCH

Based on *CityData* and *CitySimu*, we design a multi-modal urban evaluation benchmark *CityBench* to evaluate the capability of LLMs and VLMs. In the following section, we first summarize the whole pipeline and then give introduction to each task. Due to the limits of space, detailed templates and prompts for each task can refer to the appendix.

### 3.3.1 PIPELINE

Figure 2 describe the procedure of building evaluation benchmark. As introduced before, *CityData* works in the data collection and data processing stage and *CitySimu* works in the data processing stage. We focus on introducing the evaluation generation stage and quality control stage as follows.

In the evaluation generation stage, we use template based methods and LLMs/VLMs based methods to generate the evaluation questions. For example, for the image geolocalization task, the groundth location is already known when collecting, thus we directly design template based question to convert the image geolocalization task into question answer pair. As for the outdoor navigation task, we employ VLM to act as human annotation experts to annotate the data to generate the navigation instruction with additional inputs. In *CityBench*, the instructions for urban exploration task and the outdoor navigation task are generated by LLM assisted methods. Instructions for other tasks are generated from template based methods.

Due to the hand-craft designs and potential issues of LLMs, we apply a quality control stage to filter and rewrite the generated questions to obtain a high quality evaluation questions. For questions generated from template based methods, we use LLM as data quality expert to filter the low-quality data and use LLM as data rewritter to rewrite the questions with diverse formats and expressions. For questions generated from the LLMs/VLMs based methods, we use LLM/VLM as the agent with additional information to execute the task to verify the quality of generated instructions. If

the generated questions are filtered too much, we will return to the evaluation generation stage to generate new questions again. Finally, the authors of this paper also participate in the quality control stage to filter and rewrite the generated data to ensure the quality of whole benchmark.

After the above stages, we produce the evaluation benchmark with 8 urban tasks. Their relations are presented in Figure 4. Details of each task are introduced as follows.

### 3.3.2 PERCEPTION AND UNDERSTANDING TASK

**Image Geolocalization** Image geolocalization task is to predict the precise location of image based on its context. Street view image is regarded as the recording of urban appearance and play an important role in understanding the urban environment and dynamics Fan et al. (2023). Thus, we query VLMs with street view image and require them to directly generate the location of image. A good VLM should recognize the important objects from the street image and mapping them into the potential locations. Following Haas et al. (2023), we define two subtasks for this task: city name inference and precise latitude and longitude coordinates inference.

**Geospatial Prediction** Geospatial predictions are important for understanding the global sustainable development especially for developing countries, e.g., poverty estimation Jean et al. (2016) and population density estimation Tatem (2017). One of the most widely used solutions

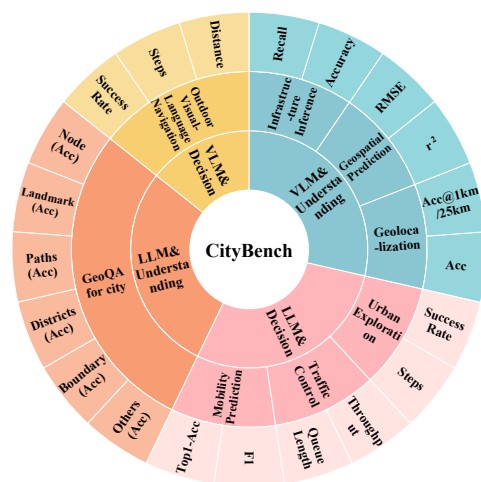

Figure 4: 8 tasks in *CityBench* with their metrics.

is using satellite images with machine learning methods to predict these socioeconomic indicators. In the benchmark, following setting from Manvi et al. (2023), we query VLMs with a satellite image as context to predict the population density of it. We use population from Worldpop Tatem (2017) as the groundtruth.

**Infrastructure Inference** Besides, we also introduce the infrastructure inference task which means to recognize the urban infrastructures from the satellite images. The groundtruth of this task is extracted from the OSM by matching predefined infrastructure key words within a fixed spatial range. Given the satellite image and a list of all kinds of infrastructures, VLM is required to generate the infrastructure names appeared in the image.

**GeoQA for City Elements** Beyond understanding the urban space from the visual perspective, we introduce geographic question answer(GeoQA) Mai et al. (2021) to test whether LLMs comprehends the fundamental spatial elements Lynch (1964) in a city from the concept view, such as road, landmarks and boundary. Following Lynch (1964); Feng et al. (2024a), we classify the spatial elements into six groups and design problems for each group.

### 3.3.3 PLANNING AND DECISION MAKING TASK

Different from the static evaluation introduced in the last section, we design four interactive decision making tasks to evaluate the capabilities of LLMs in dynamic and partial observed environments which are more challenging and realistic. With the interaction with the *CitySimu* and dynamic human activities, LLMs need to understand the important mechanisms and regularity in the urban environments to complete the decision-making tasks.

**Mobility Prediction** As one of the fundamental task for understanding the human behaviors and urban dynamics, mobility prediction task is to predict the next location of user in the next time window with given the past mobility trajectory. Here we use the the global Foursquare checkin data to support the mobility prediction in the simulator. We follow Wang et al. (2023b) to conduct the mobility prediction task via LLMs.

Table 1: Multi-source data for 13 cities around the world. The spatial coverage of these cities can be found in the appendix.

| World | Cities | Visual Data | | GeoSpatial Data | | Human Activity Data | |
|---|---|---|---|---|---|---|---|
| | | Satellite Image | Street View Image | Roads | PoI/AoIs | OD flow (>10) | Checkins |
| Asia | Beijing | 1764 | 7482 | 17043 | 276090 | 1905025 | 21015 |
| | Shanghai | 5925 | 4170 | 33321 | 57731 | 845188 | 33129 |
| | Mumbai | 638 | 6025 | 6296 | 60245 | 309147 | 31521 |
| | Tokyo | 1120 | 5514 | 33174 | 1146094 | 969865 | 1044809 |
| Europe | London | 1710 | 4148 | 14418 | 83892 | 1401404 | 173268 |
| | Paris | 238 | 6044 | 4443 | 21950 | 28362 | 85679 |
| | Moscow | 1558 | 5761 | 9850 | 28289 | 979064 | 836313 |
| Americas | NewYork | 320 | 3934 | 5414 | 349348 | 71705 | 390934 |
| | SanFrancisco | 345 | 4473 | 4171 | 73777 | 61367 | 100249 |
| | SaoPaulo | 1332 | 5184 | 28714 | 1681735 | 311830 | 808754 |
| Africa | Nairobi | 336 | 5987 | 2972 | 264101 | 135332 | 25727 |
| | CapeTown | 896 | 5175 | 5947 | 151711 | 525578 | 11591 |
| Oceania | Sydney | 1935 | 5087 | 21390 | 141997 | 438763 | 54170 |

**Outdoor Navigation** Outdoor navigation task is widely used in neurocognitive science Epstein et al. (2017) as the important benchmark for evaluating the spatial cognition of human and models. As one of the most widely-used settings in outdoor navigation task, visual-language navigation task Schumann et al. (2024); Yang et al. (2024) requires the model to follow the human-annotated language instruction to arrive to the destination with the nearby street view images as additional input. This task requires the VLMs to acquire the ability of urban visual scene understanding, language understanding and decision-making.

**Urban Exploration** Here, we define a text based street exploration task to evaluate the zero-shot navigation capability of LLMs in a new city without visual input and instructions. Different from the visual language navigation which require model to follow the language instruction and understand the scene via street view image, our urban exploration task require model to explore the region via the local information (e.g. accessed road names) provided by the simulator during action and its intrinsic knowledge of the whole urban space in the city.

**Traffic Signal Control** Traffic signal control task is one of the widely studied realistic urban decision making task in recent years Wei et al. (2019). It is challenging for existing methods due to the dynamic traffics and the generalization issues. It is to generate the future traffic signal schedule by considering the current traffic states and the future traffics. Lai et al. Lai et al. (2023) propose LLMLight to solve the single traffic signal control in simple intersection and demonstrate the generalization of LLMs. Follow this work, we extend it into multiple traffic signal control in a region with more than one intersections, which are more challenging and require cooperation between different intersections.

## 4 BENCHMARK AND EXPERIMENTS

### 4.1 SETTINGS

**Model Deployment** To facilitate usage of *CityBench*, we have implemented local deployment support for the majority of LLMs and VLMs using VLMEvalKit Duan et al. (2024) and vLLM Kwon et al. (2023). Additionally, we also support evaluation through the APIs of commercial closed-source models, e.g., OpenAI and commercial open-source models, e.g. DeepInfra [4].

**Baselines** We select well-known LLMs and VLMs as baselines. For VLMs, we select LLaVa-NeXT Liu et al. (2024c), CogVLM-v2 Wang et al. (2023a), MiniCPM-LLama3-V-2.5 OpenBMB (2024), Qwen-VL-plus and GPT4o. For LLMs, we select LLama3-8B, LLama3-70B, Mistral-7B-v0.2 Jiang et al. (2023), Mixtral-8x22B-v0.1 Jiang et al. (2024), DeepSeekv2 Shao et al. (2024), GPT3.5, and GPT4 Achiam et al. (2023). We also select representative baselines, including Geo-CLIP Vivanco Cepeda et al. (2024) for street view image geolocalization, RSVA Wang et al. (2022) for infrastructure inference, RemoteCLIP Yeh et al. (2020); Liu et al. (2024b) for population prediction, LSTPM Sun et al. (2020) for mobility prediction and MaxPressure Varaiya (2013) for traffic signal control task.

---

[4] https://deepinfra.com/

Table 2: Performance of 16 widely-used VLMs on four urban visual tasks in *CityBench*, where 'City Infer' and 'Loc Infer' denote the city inference task and the geo-coordinates inference task of street view image, 'Population' denotes the geospatial prediction task, 'Infra' denotes the infrastructure inference task, 'Navigation' denotes the outdoor visual-language navigation task.

| Tasks | Perception & Understanding | | | | Infra | Decision-making | |
| | City Infer | Loc Infer | Population | | | Navigation | |
| Metrics | Acc↑ | Acc@25km↑ | RMSE↓ | $r^2$↑ | Acc↑ | Succ Rate↑ | Distance↓ |
|---|---|---|---|---|---|---|---|
| **GeoCLIP** | 0.340 | 0.464 | - | - | - | - | - |
| **RSVA** | - | - | - | - | 0.655 | - | - |
| **RemoteCLIP** | - | - | 1.966 | 0.368 | - | - | - |
| **Qwen2VL-2B** | 0.630 | 0.407 | 2.478 | 0.008 | 0.657 | 0.020 | 679.333 |
| **InternVL2-2B** | 0.238 | 0.380 | 3.142 | -0.841 | 0.738 | 0.247 | 236.088 |
| **InternVL2-4B** | 0.398 | 0.397 | 2.501 | -0.144 | 0.735 | 0.260 | 272.445 |
| **Yi-VL-6B** | 0.000 | 0.105 | 5.471 | -3.967 | 0.816 | **0.267** | 429.683 |
| **Qwen2VL-7B** | 0.688 | 0.522 | 2.637 | -0.112 | 0.773 | 0.153 | 529.549 |
| **LLaVA-NeXT-8B** | 0.267 | 0.221 | 3.31 | -0.764 | 0.796 | 0.207 | 361.647 |
| **MiniCPM-V2.5-8B** | 0.262 | 0.223 | 3.57 | -1.054 | 0.806 | 0.260 | 296.427 |
| **InternVL2-8B** | 0.522 | 0.728 | 2.806 | -0.320 | 0.806 | 0.233 | **223.971** |
| **GLM-4v-9B** | 0.726 | 0.000 | 2.769 | -0.516 | **0.857** | 0.247 | 444.793 |
| **CogVLM2-19B** | 0.559 | 0.326 | 2.75 | -0.301 | 0.726 | 0.087 | 596.056 |
| **InternVL2-26B** | 0.429 | 0.003 | 2.683 | -0.209 | 0.790 | 0.180 | 526.079 |
| **Yi-VL-34B** | 0.251 | 0.003 | 2.510 | -0.052 | 0.790 | 0.253 | 384.005 |
| **LLaVA-NeXT-34B** | 0.501 | 0.408 | 2.61 | -0.163 | 0.804 | **0.267** | 274.036 |
| **InternVL2-40B** | 0.574 | 0.555 | 2.514 | -0.113 | 0.808 | 0.213 | 364.032 |
| **Qwen-VL-plus** | 0.793 | 0.645 | 3.14 | -1.028 | 0.454 | 0.240 | 377.622 |
| **GPT4o** | **0.862** | **0.797** | 2.32 | 0.122 | 0.812 | 0.180 | 388.582 |

**Evaluation Metrics** We follow the common practice of each task to define the metrics. Metrics and instances for each task are presented in Figure 4 and Table 6 in appendix. For each task with results from 13 cities, we report the mean value value of them in Table 2 and Table 3. More detailed results like standard deviation value can be found in the appendix.

## 4.2 EXPERIMENTAL RESULTS

**Performance of VLMs on Urban Visual Tasks** Main results on urban visual tasks are presented in Table 2. We find three interesting observations: 1) VLMs perform much better than the traditional baselines on the urban visual task, e.g, city inference via street view image and infrastructure inference, which require commonsense and semantic understanding ability of models. 2) But, VLMs perform poorly on tasks requiring precise numerical values, or have a diminished advantage, e.g., the geospatial prediction task on population and precise geo-coordindates prediction task. 3) Most of the VLMs' performance on all the tasks are not robust, e.g., some VLMs' performance is close to zero and large model within the same model series not always perform better than the smaller models, e.g., InternVL2-26B performs worse than InternVL2-8B. Finally, we find that the performance variability across different LLMs and VLMs is primarily influenced by the capabilities of the LLMs backbone. For example, in widely used VLMs for urban tasks, Intern2.5-7B consistently outperforms Qwen2-7B and Mistral-7B in most tasks, which can be attributed to Intern2.5-7B's superior performance in general NLP tasks at the same parameter scale. Similarly, LLaVA-NeXT-8B demonstrates performance comparable to MiniCPM-V2.5-8B, as both models leverage the same LLM backbone, LLaMA3-8B.

**Performance of LLMs on Urban Tasks without Visual Input** Results on urban task without visual input are presented in Table 3. Compared with the simple urban visual tasks, these urban tasks require more reasoning abilities and professional knowledge of LLMs. From Table 3, we find that LLMs perform much better than the traditional methods in mobility prediction task which is consistent with the results from Wang et al. Wang et al. (2023b). Besides, different from the results from Lai et al. Lai et al. (2023), the performance of LLMs on the traffic signal control task seems poor which are far worse than the classic Max-Pressure method. It is noted that our testing scene is the more challenging multi-intersection signal control problem. Finally, as for the GeoQA task for city elements, we find that even the best LLM GPT4-Turbo still has a long way from the performance ceiling. Additionally, we present results from several VLMs for these tasks at the bottom of Table 3. Compared to their original LLM backbones, the performance of VLMs shows some decline. For

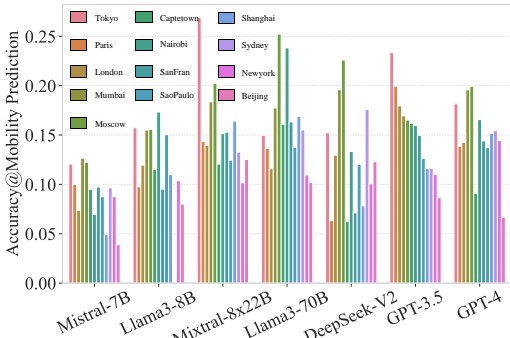 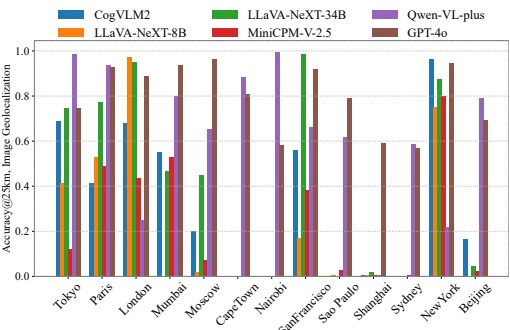

Figure 5: Detailed results of LLMs on mobility prediction task(left) for different cities and image geolocalization task(right) for different models.

instance, CogVLM2-19B performs worse than Llama3-8B in both mobility prediction and urban exploration tasks. Therefore, maintaining the general capabilities of LLMs during the training of VLMs should be a key direction to ensure their effectiveness across a wide range of tasks.

**Error Analysis of LLMs** We find that LLMs often display errors such as logic error, format error, invalid action, refusal to answer, and hallucinations. As shown in Figure 6, Llama3-8B exhibited a logical error in its judgment of time in the task mobility prediction. For the urban exploration task, Qwen2-7B refused to choose the option and instead demanded the user to use a navigation service to solve the problem. Intern2.5-7B directly stated that it lacks expertise in this area and needs more information to answer the question. Llama3-8B provided an invalid option in the traffic signal task, rendering *CitySimu* unable to perform next action. We notice that the most frequent error is Misformatted, and several instances close to 0 in Table 3 are mostly caused by formatting errors. Thus, one of the promising direction is to reduce these error from LLMs to improve their practicality.

**Error Analysis of VLMs** Urban visual tasks require the model to make decisions directly without going through an explanation process. As a result, as shown in Figure 7, common errors include format errors, invalid actions, refusal to answer and hallucinations. In the image geolocalization task, InternVL2-2B provides a response that do not follow the required format, while Yi-VL-34B gives an irrelevant invalid response. CogVLM2-19B and Yi-VL-34B, in the geospatial prediction and infrastructure inference tasks respectively, repeat the examples provided in the question and refuse to answer the actual question. Due to the response format requirements of the tasks, the most common error made by VLMs in urban visual tasks is misformatted responses.

**Geospatial Bias of LLMs and VLMs** To further investigate the difference between LLMs and VLMs, we report the detailed results of mobility prediction task and image geolocalization tasks from 13 cities in Figure 5. Based on the above results, we have made several interesting discoveries. First, we find that the performance of different LLMs varies a lot across different cities, no LLM can always perform best in mobility prediction tasks. Second, we find that the performance of VLMs on visual task like image geolocalization task are significantly biased. Most VLMs perform well in major international cities, but poorly in some lesser-known cities (e.g., CapeTown and Nairobi). We provide simple evidence for this phenomenon by analyzing the number of public websites available on Google and Wikipedia. For more details, see Section A.3 in the appendix. The variability in evaluation results demonstrate the necessity of establishing a global evaluation benchmark, and also highlights the potential shortcomings and areas for improvement of LLMs.

## 5 CONCLUSION

In this paper, we propose *CityBench*, a systematic evaluation benchmark for LLMs and VLMs in diverse urban tasks. With the data support from *CityData* and simulation support from *CitySimu*, we design 8 important urban tasks in 13 cities to constitute the *CityBench* for evaluating the capabilities of LLMs and VLMs. Extensive experiments present that LLMs and VLMs exhibit exceptional performance in various urban tasks requiring commonsense and semantic understanding, but fail in challenging urban tasks which require professional domain knowledge and precise numeric calculations. The extensive results from *CityBench* demonstrate the potential the applying LLMs and VLMs in various urban tasks and also shed light for the future research of developing more powerful LLMs and VLMs for urban tasks.

Table 3: Performance of LLMs and VLMs on four urban tasks without visual input in *CityBench*, where 'Top 1 Acc' denotes the 'Top1 Accuracy' metric, 'Queue' denotes the 'Queue Length' metric.

| Tasks | | Understanding | Planning & Decision-making | | | | | |
| | | GeoQA | Mobility Prediction | | Urban Exploration | | Traffic Signal | |
| Metrics | | Accuracy↑ | Top1 Acc↑ | F1↑ | Succ Rate↑ | Steps↓ | Queue↓ | Throughput↑ |
|---|---|---|---|---|---|---|---|---|
| Baselines | LSTPM | - | 0.114 | 0.086 | - | - | - | - |
| | Fixed-Time | - | - | - | - | - | 57.870 | 993.333 |
| | Max-Pressure | - | - | - | - | - | **36.898** | **1345.333** |
| LLMs | Mistral-7B | 0.229 | 0.090 | 0.087 | 0.730 | 5.382 | 64.120 | 853.333 |
| | Qwen2-7B | 0.289 | 0.142 | 0.109 | 0.697 | 5.889 | 62.271 | 880.000 |
| | Intern2.5-7B | 0.304 | 0.118 | 0.102 | 0.738 | 5.552 | 55.121 | 1047.667 |
| | LLama3-8B | 0.297 | 0.130 | 0.094 | 0.747 | 5.304 | 57.738 | 1014.333 |
| | Gemma2-9B | 0.339 | 0.131 | 0.120 | 0.716 | 5.679 | 74.475 | 651.333 |
| | Intern2.5-20B | 0.315 | 0.116 | 0.098 | 0.679 | 6.243 | 61.229 | 958.667 |
| | Gemma2-27B | 0.349 | 0.145 | 0.118 | 0.713 | 5.733 | 56.081 | 1009.333 |
| | Qwen2-72B | 0.357 | 0.155 | 0.135 | 0.697 | 5.887 | 66.924 | 793.333 |
| | LLama3-70B | 0.329 | **0.159** | 0.130 | **0.796** | **4.941** | 59.338 | 959.667 |
| | Mixtral-8x22B | 0.321 | 0.155 | **0.136** | 0.745 | 5.339 | 65.682 | 821.333 |
| | DeepSeekV2 | 0.358 | 0.126 | 0.101 | 0.698 | 5.739 | 56.086 | 1020.333 |
| VLMs | InternVL2-2B | 0.296 | 0.000 | 0.000 | 0.672 | 6.015 | 55.725 | 1012.000 |
| | InternVL2-4B | 0.304 | 0.130 | 0.102 | 0.674 | 6.091 | 74.499 | 647.667 |
| | InternVL2-8B | 0.329 | 0.142 | 0.102 | 0.703 | 5.714 | 53.196 | 1069.667 |
| | InternVL2-26B | 0.310 | 0.137 | 0.107 | 0.694 | 5.723 | 57.512 | 971.667 |
| | Qwen2VL-2B | 0.293 | 0.103 | 0.075 | 0.643 | 6.315 | 56.097 | 1003.667 |
| | Qwen2VL-7B | 0.286 | 0.144 | 0.102 | 0.660 | 6.155 | 55.885 | 995.333 |
| | MiniCPM-V2.5-8B | 0.308 | 0.124 | 0.092 | 0.708 | 5.643 | 56.066 | 1001.000 |
| | GLM-4v-9B | 0.296 | 0.133 | 0.092 | 0.680 | 5.979 | 53.870 | 1058.000 |
| | CogVLM2-19B | 0.282 | 0.026 | 0.029 | 0.710 | 5.905 | 55.229 | 1046.667 |
| | GPT3.5-Turbo | 0.285 | 0.152 | 0.113 | 0.719 | 5.473 | 56.219 | 1022.000 |
| | GPT4-Turbo | **0.398** | 0.147 | 0.125 | 0.757 | 5.184 | 55.761 | 1022.333 |

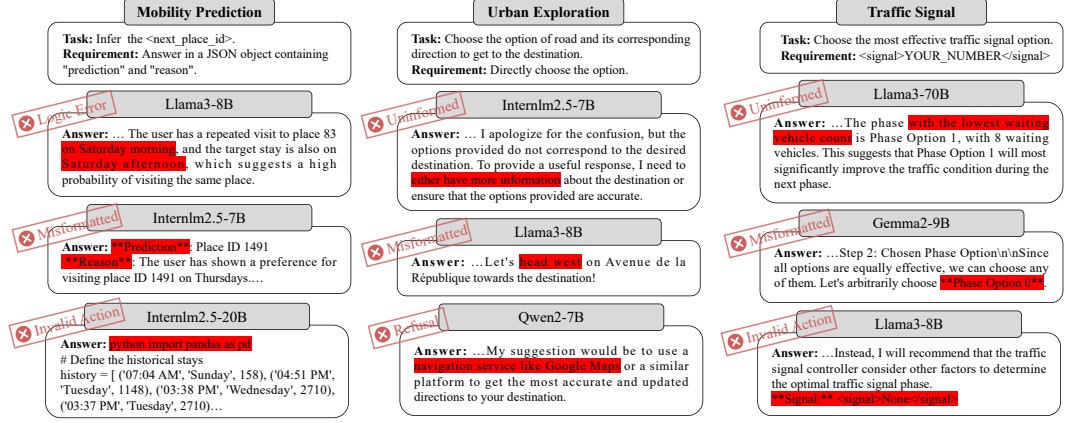

Figure 6: Error case analysis in mobility prediction, urban exploration and traffic signal control tasks.

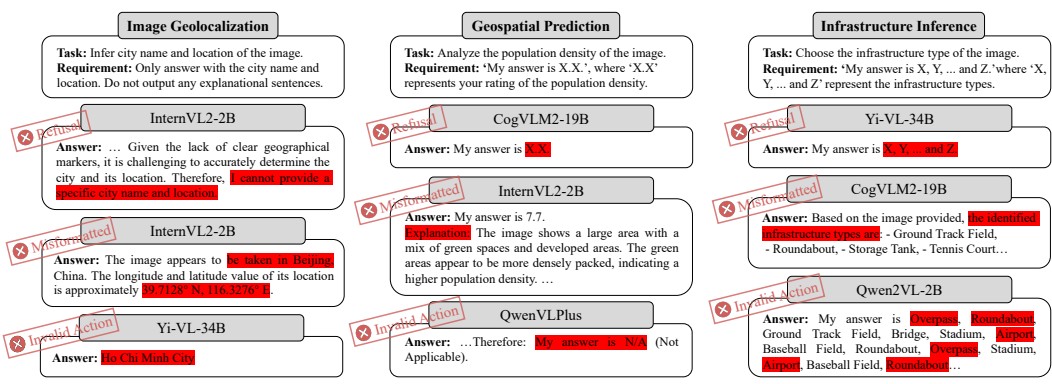

Figure 7: Error analysis in image geolocalization, geospatial prediction and infrastructure inference tasks.

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

# A APPENDIX

## A.1 FEW-SHOT PERFORMANCE OF LLMS IN CITYBENCH

Here, we present the few-shot performance of several representative LLMs in Beijing in the Table 4. For all text-based tasks, we use 2-shot as the default few-shot method. As shown in the table, the impact of few-shot learning varies across different models and tasks. For instance, few-shot learning improves performance for Gemma-27B in GeoQA but reduces it for Gemma2-9B on the same task. Similarly, it benefits Gemma2 in traffic signal tasks but proves detrimental for Llama3.

Table 4: Few-shot performance of several representative LLMs in Beijing.

| Model@Beijing | GeoQA | Mobility Prediction | | Urban Exploration | | Traffic Signal | | |
|---|---|---|---|---|---|---|---|---|
| | Accuracy | Top1 Acc | F1 | Succ Rate | Steps | Throughput | Travel Time | Queue Length |
| Gemma2-9B-fewshot | 0.306 | 0.115 | 0.101 | 0.728 | 6.152 | 1528 | 2129.21 | 64.269 |
| Gemma2-9B-zeroshot | 0.339 | 0.131 | 0.120 | 0.716 | 5.679 | 1448 | 2214.451 | 71.729 |
| Gemma2-27B-fewshot | 0.359 | 0.109 | 0.076 | 0.696 | 6.260 | 2240 | 1746.89 | 31.683 |
| Gemma2-27B-zeroshot | 0.349 | 0.145 | 0.118 | 0.713 | 5.733 | 2187 | 1762.182 | 33.322 |
| LLama3-8B-fewshot | 0.288 | 0.095 | 0.0546 | 0.692 | 6.424 | 1547 | 2132.233 | 61.84 |
| LLama3-8B-zeroshot | 0.297 | 0.130 | 0.094 | 0.747 | 5.304 | 2128 | 1873.757 | 40.941 |
| Llama3-70B-fewshot | 0.343 | 0.089 | 0.0620 | 0.796 | 4.941 | 1810 | 1962.493 | 50.541 |
| Llama3-70B-zeroshot | 0.329 | 0.159 | 0.130 | 0.74 | 5.876 | 2031 | 1893.517 | 43.475 |

## A.2 DETAILS OF QUALITY CONTROL

In CityBench, the authors will participate in the quality control process for some tasks, following the automatic quality control stage. Taking the manual checking of GeoQA as an example, the original data from OpenStreetMap contains low-quality information, with missing or incorrect details about AOI, POI, and roads. When this low-quality data is used in the evaluation task, LLMs may become confused and generate meaningless answers. In such cases, the authors review the questions to ensure that the information in the context is meaningful. However, due to time limitations for participants, we can only randomly sample the evaluation cases. For instances from cities and regions where authors are unfamiliar, we filter out low-quality instances. For instances from cities and regions where authors are familiar, we rewrite low-quality instances using external information, such as commercial map services. Finally, if data quality remains unsatisfactory after filtering and rewriting, we will regenerate a certain number of cases to fill the gaps. In fact, while considering the probabilities of filtering, we generate enough candidate instances during the initial generation.

## A.3 ADDITIONAL RESULTS OF GEOSPATIAL BIAS ANALYSIS

Taking three well-performing cities (New York, London, Paris) and three under-performing cities (Shanghai, CapeTown, Nairobi) as examples, Table 5 illustrates the relationship between the performance of street view image localization tasks and the size of the training corpus for LLMs. The training corpus size is approximated by the number of Google search entries and Wikipedia entries for each city. Compared to the well-performing cities, the under-performing cities have significantly smaller 'training corpora' in the public websites. Additionally, open-source models exhibit significantly worse performance than commercial models in terms of geospatial bias. This observation provides initial evidence for analyzing geographical bias, but we believe there are more diverse factors contributing to this phenomenon.

## A.4 MAP BUILDING TOOL IN *CityData*

The map building tool enhances open-source map data to support subsequent behavior simulations, encompassing lane topology recovery, relationship recognition, intersection reconstruction, area of interest (AOI) mapping, point of interest (POI) clustering, basic traffic rule generation, and right-of-way construction. The framework of map building tool is presented in Figure 8.

Table 5: The performance of street view image localization tasks in different cities and its relationship with the amount of data in the 'training corpus' of LLMs, where the amount of data in the training corpus is approximated by the number of Google search entries and Wikipedia entries for each city.

| Cities | GPT4o | MiniCPM-V2.5 | Google Pages | Wikipedia Pages |
|---|---|---|---|---|
| Shanghai | 59% | 0.4% | 582,000,000 | 62,495 |
| CapeTown | 81% | 0 | 587,000,000 | 54,724 |
| Nairobi | 58.19% | 0 | 205,000,000 | 14,226 |
| New York | 94.39% | 80% | 7,070,000,000 | 1,039,264 |
| London | 88.80% | 43.8% | 5,460,000,000 | 848,381 |
| Paris | 92.80% | 49% | 4,170,000,000 | 353,500 |

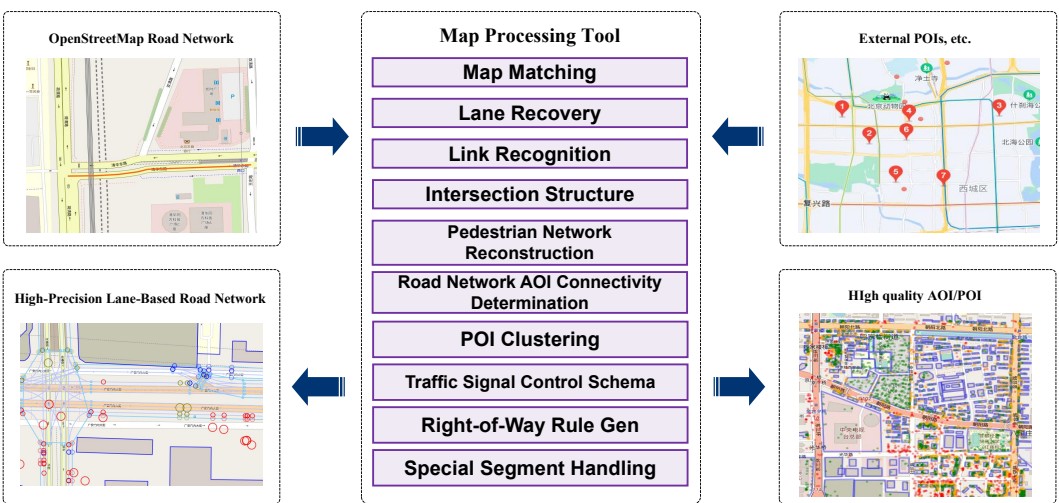

Figure 8: Framework of map building tool.

Table 6: Detailed information of 8 evaluation tasks in *CityBench*, including data modality, metric and data instances. Task settings across different cities keep consistent.

| CityBench | Tasks | Modality | Metrics | Instances | Images |
|---|---|---|---|---|---|
| **Perception&** **Understanding** | Image Geolocalization | Image | Acc, Acc@1km/25km | 6500 | 6500 |
| | Geospatial Prediction | Image | $r^2$, RMSE | 5739 | 5739 |
| | Infrastructure Inference | Image | Accuracy, Recall | 5739 | 5739 |
| | GeoQA for City Elements | Text | Accuracy | 13126 | / |
| **Planning&** **Decision Making** | Mobility Prediction | Text | Top1-Acc, F1 | 6500 | / |
| | Urban Exploration | Text | Steps, Success Rate | 650 | / |
| | Outdoor Navigation | Image | Distance, Steps, Success Rate | 650 | 55984 |
| | Traffic Signal Control | Text | Queue Length, Throughput | 1hour $\times$ 13 cities | / |

A.5 DISCUSSION

**Limitations.** While our platform is based on the public data from various sources, the quality of different data may play a important role in the evaluation results. In the future, we plan to collect more kinds of tasks with global scale groundtruth data to further improve the reliability and representativeness of benchmark.

**Ethical considerations and potential societal impact.** Our benchmark is designed for enable the global evaluation of LLMs and VLMs for various cities with different cultures and countries. We try our best to improve the ease-of-use and fairness for cities with different development levels. However, due to the limitation of accessed data, the evaluation results for different cites varies a lot. Therefore, the variation in evaluation results caused by data quality may lead to a certain degree of misunderstanding regarding the performance on some urban problems. We call the whole community for attention to this issue to improve the usability of LLMs across different races and countries, promoting fairness and sustainable development of the world.

**Develop foundation model for urban domain.** Based on the results of our benchmark, we find existing LLMs perform poorly on many urban tasks, even worse than some classic simple baseline algorithms. Developing LLMs tailored for urban domain is urgently necessary. We hope our benchmark can accelerate this development and we look forward to a more comprehensive and robust evaluation framework for urban domain.

A.6 DETAILED RESULTS FOR EACH TASK

Results of visual perception tasks of multi-modal LLMs are presented in Table 7. In general, commercial LLMs like GPT4o performs best in most tasks, achieving the best results in 5 out of 7 metrics. Among the open source models, LLaVA-NeXT-34B performs best with achieving 3 second best results in 7 metrics. Besides, we can find that the performance of open source LLM varies significantly across different cities which indicates significant geospatial bias of open source LLMs. Detailed analysis about this can refer to section 4.2. Results of geospatial knowledge QA task of LLMs are presented in Table 8. As Table 8 shows, commercial LLMs GPT4-Turbo performs best in 4 out of 6 tasks and the left two performs second best. In summary, in terms of geospatial knowledge about urban spaces, the open-source LLMs lag far behind the commercial LLMs.

Table 9 presents the results of mobility prediction task and urban exploration task, the performance difference between LLMs in decision making tasks are much smaller than the last section. It is amazing to observe that open source LLMs (e.g., LLama3-70B) performs better than commercial LLMs in these two tasks. Results of traffic signal control task are presented in Table 11. Here, we report results from three cities with various traffic volume and environment. In general, commercial LLMs like GPT-3.5 or GPT-4 performs better but small open source LLMs can also achieve competitive results in some cases. One interesting direction is to investigate these difference and find out the underlying factors behind them.

A.7 PARAMETER SETTINGS

In the experiments, the default generation parameters for LLMs are presented in Table 12. Besides, we modify some parameters for some experiments. For street navigation tasks, the temperature is changed to 0.7 for better exploration. For multi-modal LLM, the temperature of CogVLM2 is 0.8 and the temperature of MiniCPM-V-2.5 is 0.7.

A.8 DISTRIBUTION OF SATELLITE AND STREET VIEW IMAGES IN 13 CITIES

The spatial distribution of sampled satellite and street view images in 13 cities are presented in Figure 9. In each subfigure, the top figure shows the spatial distribution of sampled satellite images, with blue points indicating the actual collection locations and red points representing the sampled locations in the benchmark. Similarly, the bottom figure shows the distribution of sampled street view images, with blue points for collected images and red points for sampled ones.

Table 7: Results of VLMs on three representative visual understanding tasks. For most metrics, higher values are better, except for RMSE. In the table, **bold** denotes the best results, underline denotes the second best results.

| Models | Image Geolocalization | | | Population Density | | Infrastructures | |
|---|---|---|---|---|---|---|---|
| | City Accuracy | Accuracy @1km | Accuracy @25km | $r^2$ | RMSE | Precision | Accuracy |
| GeoCLIP | 0.340±0.195 | **0.0254±0.0498** | 0.4643±0.222 | - | - | - | - |
| RemoteCLIP | - | - | - | - | - | 0.126±0.20 | 0.655±0.226 |
| LSMSDelta | - | - | - | **0.368±0.416** | 1.966±0.678 | - | - |
| Qwen2VL-2B | 0.630±0.411 | 0.0123±0.0279 | 0.407±0.380 | 0.008±0.303 | 2.478±0.603 | 0.101±0.176 | 0.657±0.295 |
| InternVL2-2B | 0.238±0.293 | 0.008±0.023 | 0.380±0.381 | -0.841±1.421 | 3.142±0.492 | 0.099±0.194 | 0.738±0.236 |
| InternVL2-4B | 0.398±0.383 | 0.0126±0.0357 | 0.397±0.427 | -0.144±0.840 | 2.501±0.532 | 0.131±0.241 | 0.735±0.253 |
| Yi-VL-6B | 0.000 | 0.0013±0.0044 | 0.1050±0.265 | -3.967±2.142 | 5.471±1.22 | 0.151±0.216 | 0.816± 0.166 |
| Qwen2VL-7B | 0.688±0.420 | 0.011±0.027 | 0.522±0.463 | -0.112±0.428 | 2.637±0.800 | 0.145±0.221 | 0.773±0.215 |
| LLaVA-NeXT-8B | 0.267±0.322 | 0.0097±0.0269 | 0.221±0.337 | -0.764±0.582 | 3.31±0.823 | 0.245±0.301 | 0.796±0.187 |
| MiniCPM-V2.5-8B | 0.262±0.267 | 0.0061±0.0124 | 0.223±0.270 | -1.054±0.740 | 3.57±0.621 | 0.251±0.323 | 0.806±0.197 |
| InternVL2-8B | 0.522±0.434 | 0.032±0.0574 | 0.728±0.247 | -0.320±0.596 | 2.806±0.68 | 0.135±0.217 | 0.806±0.187 |
| GLM-4v-9B | 0.726±0.416 | 0.000 | 0.000 | -0.516±1.366 | 2.769±0.614 | 0.017±0.132 | **0.857±0.231** |
| CogVLM2-19B | 0.559±0.333 | 0.0067±0.0176 | 0.326±0.335 | -0.301±0.737 | 2.75±0.504 | **0.287±0.320** | 0.726±0.262 |
| InternVL2-26B | 0.429±0.385 | 0.001±0.002 | 0.003±0.007 | -0.209±0.546 | 2.683±0.626 | 0.124±0.216 | 0.790±0.178 |
| Yi-VL-34B | 0.251±0.281 | 0.0006±0.002 | 0.003±0.007 | -0.052±0.487 | 2.510±0.697 | 0.133±0.219 | 0.790±0.182 |
| LLaVA-NeXT-34B | 0.501±0.387 | 0.0123±0.0328 | 0.408±0.413 | -0.163±0.603 | 2.61±0.643 | 0.262±0.297 | 0.804±0.187 |
| InternVL2-40B | 0.574±0.447 | 0.013±0.034 | 0.555±0.481 | -0.113±0.689 | 2.514±0.604 | 0.124±0.206 | 0.808±0.161 |
| Qwen-VL-plus | 0.793±0.288 | 0.0052±0.0039 | 0.645±0.313 | -1.028±1.667 | 3.14±1.016 | 0.203±0.233 | 0.454±0.170 |
| GPT4o | **0.862±0.079** | 0.0187±0.0375 | **0.797±0.148** | 0.122±0.309 | **2.32±0.664** | 0.248±0.306 | 0.812±0.195 |

Table 8: Results of different LLMs on GeoQA for city, a geospatial knowledge question answering task for urban space. Here, the higher value is better, **bold** denotes the best results, underline denotes the second best results.

| Models | Nodes | Landmarks | Paths | Districts | Boundary | Others |
|---|---|---|---|---|---|---|
| Mistral-7B | 0.2286±0.0167 | 0.1950±0.0083 | 0.2054±0.0179 | 0.1511±0.0150 | 0.3269±0.0232 | 0.2688±0.0186 |
| Qwen2-7B | 0.2214±0.0174 | 0.2762±0.0121 | 0.2709±0.0129 | 0.2906±0.0371 | 0.3208±0.0118 | 0.3568±0.0157 |
| Intern2.5-7B | 0.2035±0.0096 | 0.2946±0.0167 | 0.2783±0.0142 | 0.2897±0.0451 | 0.4296±0.0447 | 0.3296±0.0183 |
| LLama3-8B | 0.2322±0.0150 | 0.2765±0.0143 | 0.3173±0.0054 | 0.2627±0.0163 | 0.3569±0.0109 | 0.3365±0.0225 |
| Gemma2-9B | 0.2605±0.0180 | **0.3988±0.0443** | 0.2915±0.0109 | 0.3277±0.0571 | 0.3738±0.0336 | **0.3809±0.0223** |
| Intern2.5-20B | 0.2371±0.0117 | 0.3792±0.0409 | 0.3422±0.0080 | 0.2280±0.0279 | 0.3808±0.0329 | 0.3249±0.0222 |
| Gemma2-27B | 0.2567±0.0235 | 0.3569±0.0308 | 0.3608±0.0112 | **0.3734±0.0279** | 0.3781±0.0229 | 0.3715±0.0177 |
| Qwen2-72B | 0.3047±0.0247 | 0.3758±0.0376 | 0.3849±0.0399 | 0.3146±0.0541 | 0.4135±0.0135 | 0.3521±0.0220 |
| LLama3-70B | 0.2646±0.0076 | 0.3304±0.0320 | 0.4048±0.0334 | 0.2532±0.0429 | 0.3727±0.0112 | 0.3485±0.0185 |
| Mixtral-8x22B | 0.2633±0.0155 | 0.3569±0.0367 | 0.3652±0.0201 | 0.2730±0.0370 | 0.3415±0.0125 | 0.3239±0.0241 |
| DeepSeekV2 | 0.2904±0.0065 | 0.3035±0.0165 | 0.4133±0.0208 | 0.3518±0.0399 | **0.4650±0.0363** | 0.3264±0.0232 |
| GPT3.5-Turbo | 0.2477±0.0260 | 0.2627±0.0128 | 0.3222±0.0255 | 0.1909±0.0244 | 0.3788±0.0137 | 0.3050±0.0244 |
| GPT4-Turbo | **0.3429±0.0095** | 0.3938±0.0393 | **0.5249±0.0349** | 0.3054±0.0412 | 0.4485±0.0198 | 0.3707±0.0219 |

Table 9: Results of LLMs on mobility prediction and urban exploration tasks. The arrow next to the metric indicates the direction of improvement, an upward arrow indicates that higher values of the metric are better.

| Models | Mobility Prediction | | Urban Exploration | |
|---|---|---|---|---|
| | Top1 Accuracy ↑ | F1 ↑ | Success Rate ↑ | Average Steps ↓ |
| LSTPM | 0.114±0.0013 | 0.086±0.001 | - | - |
| Mistral-7B | 0.090±0.0255 | 0.087±0.031 | 0.730±0.134 | 5.382±1.982 |
| Qwen2-7B | 0.142±0.0328 | 0.109±0.028 | 0.697±0.076 | 5.889±1.180 |
| Intern2.5-7B | 0.118±0.0361 | 0.102±0.033 | 0.738±0.064 | 5.552±0.947 |
| LLama3-8B | 0.130±0.0320 | 0.094±0.030 | 0.747±0.140 | 5.304±2.059 |
| Gemma2-9B | 0.131±0.0428 | 0.120±0.039 | 0.716±0.079 | 5.679±1.154 |
| Intern2.5-20B | 0.116±0.0379 | 0.098±0.031 | 0.679±0.077 | 6.243±1.149 |
| Gemma2-27B | 0.145±0.0282 | 0.118±0.028 | 0.713±0.092 | 5.733±1.295 |
| Qwen2-72B | 0.155±0.0317 | 0.135±0.032 | 0.697±0.073 | 5.887±1.097 |
| LLama3-70B | **0.159±0.0430** | 0.130±0.034 | **0.796±0.085** | **4.941±1.592** |
| Mixtral-8x22B | 0.155±0.0419 | **0.136±0.042** | 0.745±0.133 | 5.339±1.961 |
| DeepSeekV2 | 0.126±0.0520 | 0.101±0.044 | 0.698±0.119 | 5.739±1.812 |
| GPT3.5-Turbo | 0.152±0.0388 | 0.113±0.035 | 0.719±0.143 | 5.473±2.015 |
| GPT4-Turbo | 0.147±0.0357 | 0.125±0.035 | 0.757±0.091 | 5.184±1.520 |

Table 10: Results of LLMs on traffic signal tasks on three cities with representative traffic volume. We used real-world mobility demands to generate traffic to support the simulation. The traffic signals we actually control in the experiment are only a part of the trajectory path, while the remaining signals are operated according to the default fixed-time scheduler.

| Methods | Paris | | | Newyork | | | Beijing | | |
| | Throughput | Travel Time | Queue Length | Throughput | Travel Time | Queue Length | Throughput | Travel Time | Queue Length |
|---|---|---|---|---|---|---|---|---|---|
| FixedTime | 296 | 2668.313 | 46.605 | 532 | 2689.421 | 90.982 | 2152 | 1799.275 | 36.023 |
| MaxPressure | **655** | **2113.981** | **24.342** | **867** | **2298.922** | **70.121** | **2514** | **1560.866** | **16.232** |
| Mistral-7B | 290 | 2656.038 | 46.795 | 574 | 2634.054 | 87.807 | 1696 | 2059.840 | 57.759 |
| Qwen2-7B | 329 | 2612.115 | 45.458 | 539 | 2640.588 | 88.757 | 1772 | 1979.836 | 52.597 |
| Intern2.5-7B | 339 | 2603.762 | 45.411 | 582 | 2601.627 | 85.473 | 2222 | 1773.167 | 34.481 |
| LLama3-8B | 335 | 2605.558 | 45.283 | 580 | 2635.365 | 86.993 | 2128 | 1873.757 | 40.941 |
| Gemma2-9B | 116 | 2766.652 | 56.236 | 390 | 2716.232 | 95.461 | 1448 | 2214.451 | 71.729 |
| Intern2.5-20B | 328 | 2612.745 | 45.838 | 488 | 2675.345 | 95.357 | 2060 | 1853.243 | 42.493 |
| Gemma2-27B | 334 | 2609.845 | 45.209 | 507 | 2674.032 | 89.712 | 2187 | 1762.182 | 33.322 |
| Qwen2-72B | 332 | 2607.622 | 45.293 | 528 | 2664.426 | 89.279 | 1520 | 2115.454 | 66.201 |
| DeepSeek-67B | 331 | 2614.453 | 45.528 | 505 | 2682.665 | 89.781 | 2237 | 1765.086 | 33.575 |
| Llama3-70B | 332 | 2607.275 | 45.424 | 516 | 2664.102 | 89.116 | 2031 | 1893.517 | 43.475 |
| Mixtral-8x22B | 334 | 2614.251 | 45.473 | 526 | 2655.111 | 88.567 | 1604 | 2093.986 | 63.006 |
| DeepSeekv2 | 333 | 2607.169 | 45.574 | 495 | 2682.194 | 89.914 | 2233 | 1754.923 | 32.771 |
| GPT3.5-Turbo | 323 | 2618.222 | 45.768 | 544 | 2649.214 | 87.926 | 2199 | 1778.963 | 34.965 |
| GPT4-Turbo | 332 | 2606.985 | 45.344 | 495 | 2670.238 | 89.706 | 2240 | 1746.608 | 32.234 |

Table 11: Detailed results of 13 VLMs on outdoor visual-language navigation tasks on Paris, Newyork and Beijing.

| Models | Paris | | | Newyork | | | Beijing | | |
| | Success Rate | Average Steps | Average Distance | Success Rate | Average Steps | Average Distance | Success Rate | Average Steps | Average Distance |
|---|---|---|---|---|---|---|---|---|---|
| Qwen2VL-2B | 0.00 | 0.10 | 649.527 | 0.06 | 0.40 | 633.234 | 0.00 | 0.14 | 755.240 |
| InternVL2-2B | 0.22 | 1.84 | 270.809 | **0.34** | 2.18 | **164.118** | 0.18 | 2.40 | 273.337 |
| InternVL2-4B | 0.22 | 1.62 | 318.246 | 0.34 | 2.00 | 230.406 | 0.22 | 2.44 | 268.684 |
| Yi-VL-6B | 0.26 | 0.94 | 437.584 | 0.34 | 1.60 | 361.249 | 0.20 | 1.66 | 490.216 |
| Qwen2VL-7B | 0.12 | 0.60 | 508.745 | 0.20 | 1.12 | 450.643 | 0.14 | 1.02 | 629.261 |
| LLaVA-NeXT-8B | 0.18 | 0.86 | 468.477 | 0.32 | 1.70 | 293.783 | 0.12 | 1.72 | 322.683 |
| MiniCPM-V2.5-8B | 0.22 | 1.52 | 349.067 | 0.34 | 2.00 | 293.465 | 0.22 | 2.34 | 246.749 |
| InternVL2-8B | 0.16 | 1.90 | **243.161** | 0.34 | 2.14 | 222.562 | 0.20 | **2.68** | 206.189 |
| GLM-4v-9B | 0.18 | 0.68 | 519.489 | 0.34 | 1.46 | 406.329 | 0.22 | 1.72 | 408.560 |
| CogVLM2-19B | 0.06 | 0.30 | 605.055 | 0.10 | 0.56 | 547.929 | 0.10 | 0.88 | 635.185 |
| InternVL2-26B | 0.14 | 1.04 | 521.734 | 0.18 | 1.12 | 574.102 | 0.22 | 1.78 | 482.403 |
| Yi-VL-34B | 0.22 | 0.96 | 463.780 | 0.34 | 1.72 | 288.500 | 0.20 | 1.76 | 399.736 |
| LLaVA-NeXT-34B | 0.22 | 1.60 | 308.899 | 0.34 | 1.90 | 247.323 | 0.24 | 2.52 | 265.886 |
| InternVL2-40B | 0.22 | 1.52 | 339.362 | 0.26 | 1.40 | 453.538 | 0.16 | 2.30 | 299.197 |
| Qwen-VL-plus | 0.18 | 0.96 | 442.616 | 0.32 | 1.76 | 302.899 | 0.22 | 2.16 | 387.353 |
| GPT4o-Mini | **0.34** | **2.02** | 296.231 | 0.32 | **2.22** | 226.265 | **0.28** | 2.46 | 234.481 |
| GPT4o | 0.24 | 1.28 | 429.289 | 0.14 | 1.30 | 405.259 | 0.16 | 1.92 | 331.199 |

Table 12: Generation parameter settings for LLMs during the experiments.

| Pameters | Open Source LLM(DeepInfra) | Commercial API | Open Source MLLM | Commercial API for MLLM |
|---|---|---|---|---|
| temperature | 0 | 0 | 0.2 | 1.0 |
| top p | 1.0 | 1.0 | 1.0 | 1.0(GPT4o), 0.8(Qwen-VL-plus) |
| max tokens | 512 | 512 | 512 | 100 |
| presence penalty | 0 | 0 | 0 | 0 |
| frequency penalty | 0 | 0 | 0 | 0 |
| repetition penalty | 1 | 1 | 1 | 1 |

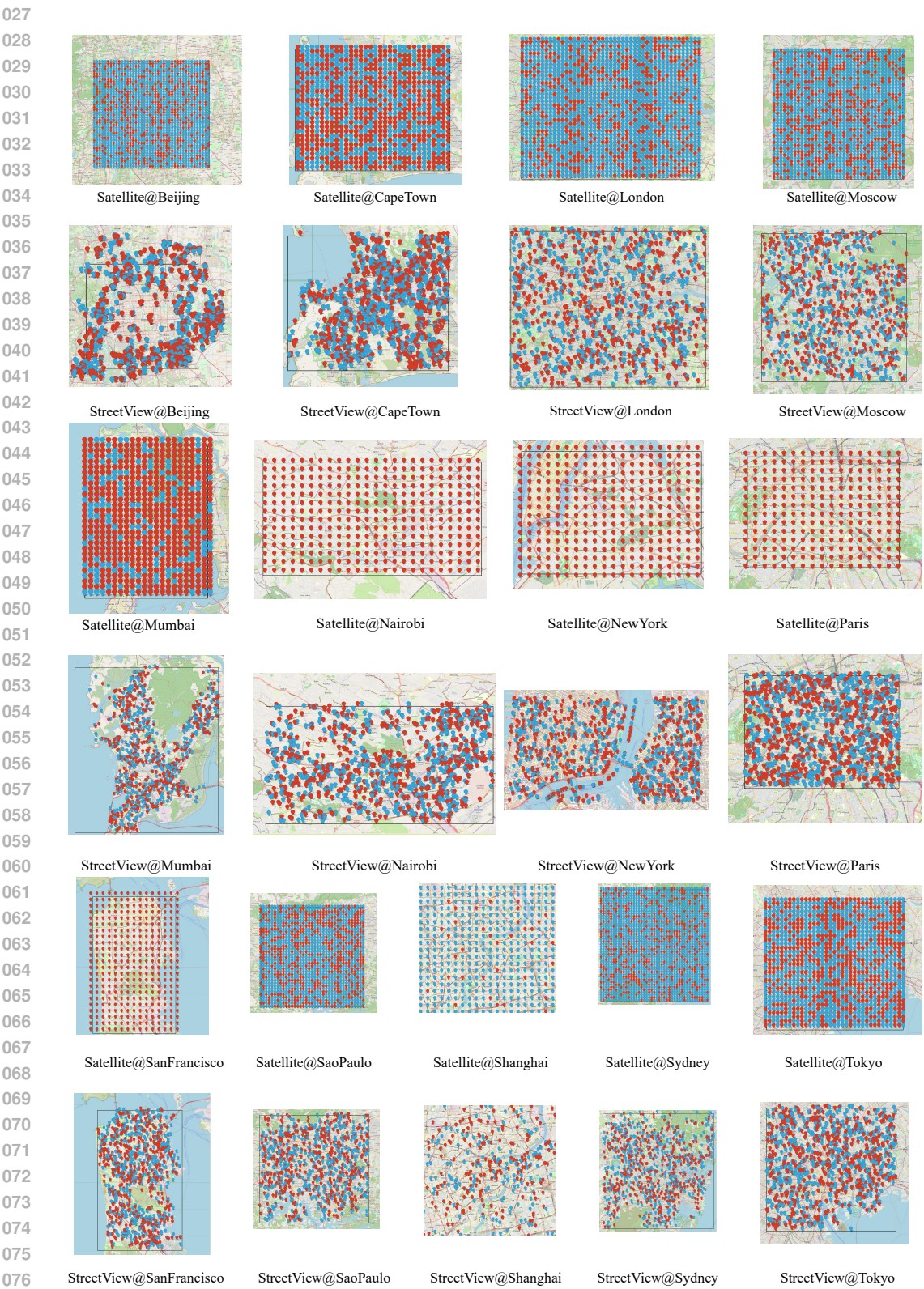

Figure 9: Distribution of satellite and street view images in 13 cities.

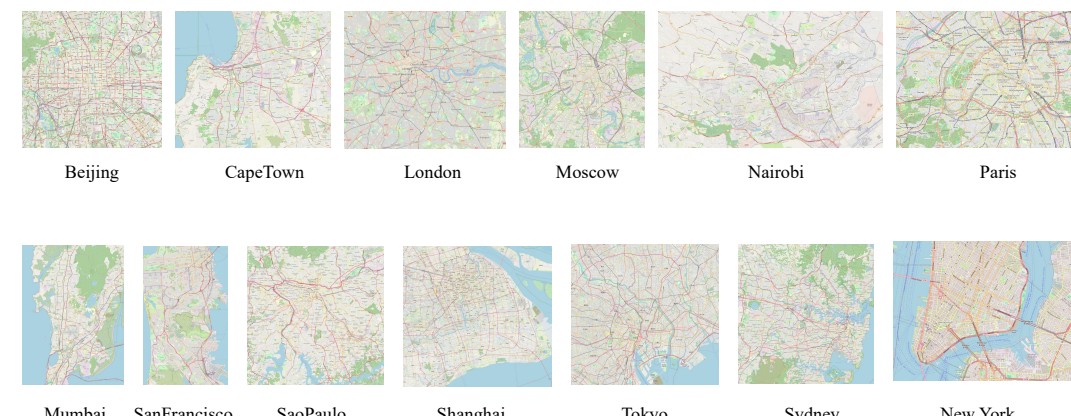

| | | | | | |
|---|---|---|---|---|---|
| Beijing | CapeTown | London | Moscow | Nairobi | Paris |
| Mumbai SanFrancisco | SaoPaulo | Shanghai | Tokyo | Sydney | New York |

Figure 10: Maps for 13 cities.

## A.9 SPATIAL RANGE OF 13 CITIES

Selected spatial range of 13 cities are presented in Figure 10.

## A.10 PROMPT EXAMPLES

**Q:** "Suppose you are an expert in geo-localization. Please first analyze which city is this image taken from, and then make a prediction of the longitude and latitude value of its location.
You can choose among: 'Los Angeles', 'Nakuru', 'Johannesburg', 'Rio de Janeiro', 'CapeTown', 'London', 'Moscow', 'Mumbai', 'Paris', 'Sao Paulo', 'Sydney', 'Tokyo', 'New York', 'Guangzhou', 'Kyoto', 'Melbourne', 'San Francisco', 'Nairobi', 'Beijing', 'Shanghai', 'Bangalore', 'Marseille', 'Manchester', 'Saint Petersburg'.
Only answer with the city name and location. Do not output any explanational sentences. Example Answer: Los Angeles. (34.148331, -118.324755)."

**A(GPT-4o):** \*\*Paris. (48.8566, 2.3522).\*\* The reason is that the architectural style, the cobblestone street, and the layout of the buildings are characteristic of Paris. The building's design, with its distinct brickwork and window shutters, is typical of Parisian residential architecture. Additionally, the road and sidewalk design are common in Paris, and the overall feel of the street is reminiscent of Parisian neighborhoods.

**A(Qwen-VL-Plus):** Paris. (48.906849, 2.35407)
**A(CogVLM-2):** Paris. (48.856522, 2.303104).
**A(LLaVA-Next-8b):** London. (51.507351, -0.127659)
**A(LLaVA-34b):** Paris. (48.856614, 2.352221)

**A(MiniCPM-v2.5):** The image provided does not contain sufficient information to accurately determine the city it was taken from. Geo-localization typically requires identifiable landmarks, signage, or distinctive architectural features that are unique to a specific city. Without such information, any attempt at prediction would be purely speculative and not based on reliable evidence. Therefore, I cannot provide a city name and location for this image.

**Street View Image in Paris**
Latitude: 48.82215588839296
Longitude: 2.35351508700578

Figure 11: Prompt example for geolocalization with street view image.

**Q:** Suppose you are an expert in identifying urban infrastructure. Please analyze this image and choose the infrastructure type from the following list: ['Bridge', 'Stadium', 'Ground Track Field', 'Baseball Field','Overpass', 'Airport', 'Golf Field', 'Storage Tank', 'Roundabout', 'Swimming Pool', 'Soccer Ball Field', 'Harbor', 'Tennis Court', 'Windmill', 'Basketball Court', 'Dam', 'Train Station'] Please meet the following requirements: 1. If you can identify multiple infrastructure types, please provide all of them. 2. You must provide the answer in the exact format: 'My answer is X, Y, ... and Z.' where 'X, Y, ... and Z' represent the infrastructure types you choose. 3. If you cannot choose any of the infrastructure types from the list, please choose 'Other'. 4. Please tell me why."

**A(CogVLM-2):** My answer is Storage Tank, Harbor, Ground Track Field, Roundabout, Overpass, and Airport.
**A(LLaVA-Next-8b):** My answer is Storage Tank, Harbor, and Other.
**A(LLaVA-34b):** My answer is Storage Tank.
**A(MiniCPM-v2.5):** My answer is Storage Tank, Ground Track Field.

**A(GPT-4o):** My answer is Harbor and Storage Tank.
**A(Qwen-VL-Plus):** My answer is Storage Tank, Train Station, Harbor, Bridge, Roundabout, Overpass, Dam, Golf Field, Windmill, Tennis Court, Soccer Ball Field, Baseball Field, Airport, Stadium, Ground Track Field, Swimming Pool, Basketball Court, Other.

**Satellite Image in London**
Harbor and Storage Tank

Figure 12: Prompt example for infrastructure inference with remote sensing image

**Q:** Please analyze the population density of the image shown comparing to all cities around the world. Rate the population density in a degree from 0.0 to 9.9, where higher rating represents higher population density. First, you should provide your answer in the exact format: 'My answer is X.X.', where 'X.X' represents your rating of population density. Then, you should give the reason.

**A(CogVLM-2):** My answer is 6.5
**A(LLaVA-Next-8b):** My answer is 0.0
**A(LLaVA-34b):** My answer is 0.1
**A(MiniCPM-v2.5):** The image provided is an aerial view of a port area, which is typically characterized by industrial and commercial structures rather than residential areas. ......Therefore, based on the visual information alone, it is not feasible to assign a numerical value to the population density of the area depicted in the image

**A(GPT-4o):** My answer is 1.0
**A(Qwen-VL-Plus):** Failed

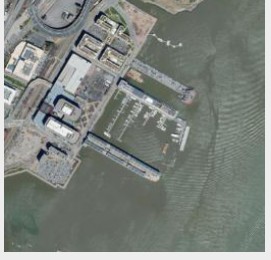

**Satellite Image in New York**
My answer is 2.7

Figure 13: Prompt example for geospatial prediction with remote sensing image.

**Q:** A traffic light controls a complex intersection with 4 signal phases. Each signal relieves vehicles' flow in the allowed lanes.

**Traffic Signal visualization**

The state of the intersection is listed below. It describes:
- The number of lanes relieving vehicles' flow under each traffic light phase.
- The total count of vehicles present on lanes permitted to move of each signal.
- The count of vehicles queuing on lanes allowed to move, which are close to the end of the road or intersection and moving at low speeds of each signal.

Available Phase Options:
  Phase Option 0: -Allowed lane counts: 12, -Vehicle counts: 24, -Waiting vehicle counts: 13
  Phase Option 1: -Allowed lane counts: 12, -Vehicle counts: 19, -Waiting vehicle counts: 8
  Phase Option 2: -Allowed lane counts: 11, -Vehicle counts: 27, -Waiting vehicle counts: 13
  Phase Option 3: -Allowed lane counts: 13, -Vehicle counts: 26, -Waiting vehicle counts: 14

Please answer:
  Which is the most effective traffic signal that will most significantly improve the traffic condition during the next phase?

Note:
  The traffic congestion is primarily dictated by the waiting vehicles, with the MOST significant impact. You MUST pay the MOST attention to lanes with long queue lengths.

Requirements:
  - Let's think step by step.
  - You can only choose one of the phase option NUMBER listed above.
  - You must follow the following steps to provide your analysis:
    Step 1: Provide your analysis for identifying the optimal traffic signal.
    Step 2: Answer your chosen phase option number.
  - Your choice can only be given after finishing the analysis.
  - Your choice must be identified by the tag: <signal>YOUR_NUMBER</signal>.

**A(GPT-4o):** …We should choose the phase option with the highest number of waiting vehicles to relieve the most congestion. Phase Option 3 has the highest number of waiting vehicles (14).
   Answer: <signal>3</signal>

**A(GPT-4-Turbo):** …From the analysis, Phase Option 3 has the highest number of waiting vehicles (14), which indicates a significant queue and potential bottleneck. This phase also has the highest number of allowed lanes (13), which suggests that activating this signal could potentially clear the congestion more effectively by allowing more vehicles to move through the intersection compared to other phases.
   Answer: <signal>3</signal>

**A(GPT-3.5-Turbo):** …Based on the waiting vehicle counts alone, Phase Option 1 has the lowest number of waiting vehicles, indicating that it may be the most effective in improving traffic flow during the next phase.
   Answer: <signal>1</signal>

**A(GPT-4o-Mini):** …The most critical factor in improving traffic conditions is to reduce the number of waiting vehicles. Phase Option 1 has the lowest number of waiting vehicles (8), which suggests it would alleviate congestion more effectively than the other options. Although it has fewer allowed lanes and vehicle counts, the significant reduction in waiting vehicles makes it the optimal choice.
   Answer: <signal>1</signal>

**A(Mixtral-8x22B):** …To determine the most effective traffic signal that will most significantly improve the traffic condition during the next phase, we need to consider the number of waiting vehicles in each phase option. As mentioned, the traffic congestion is primarily dictated by the waiting vehicles, with the most significant impact.
   Answer: <signal>1</signal>

**A(Mistral-7B):** …But of the given phase options, Phase Option 1 has the least number of waiting vehicles (8) compared to the other phase options. Therefore, choosing Phase Option 1 as the next traffic signal will most significantly improve the traffic condition during the next phase by reducing the number of waiting vehicles and allowing more vehicles to flow through the intersection.
   Answer: <signal>1</signal>

**A(Llama-3-8B):** …Phase Option 1 has the lowest waiting vehicle count, which indicates that it may be the most effective traffic signal in improving traffic conditions during the next phase.
   Answer: <signal>1</signal>

**A(Llama-3-70B):** …To determine the most effective traffic signal, we need to identify the phase that will most significantly improve the traffic condition during the next phase. Since traffic congestion is primarily dictated by waiting vehicles, we should focus on the lanes with long queue lengths. The phase with the lowest waiting vehicle count is Phase Option 1, with 8 waiting vehicles. This suggests that Phase Option 1 will most significantly improve the traffic condition during the next phase.
   Answer: <signal>1</signal>

**A(DeepSeekv2): Failed**

Figure 14: Prompt example for traffic signal control.

Q: Your task is to predict <next_place_id> in <target_stay>, a location with an unknown ID, while temporal data is available.
Predict <next_place_id> by considering:
1. The user's activity trends gleaned from <historical_stays> and the current activities from <context_stays>.
2. Temporal details (start_time and day_of_week) of the target stay, crucial for understanding activity variations.
Present your answer in a JSON object with:
"prediction" (IDs of the five most probable places, ranked by probability) and "reason" (a concise justification for your prediction).
The data:
<historical_stays>: {historical_stays}
<context_stays>: {context_stays}
<target_stay>: {target_time, <next_place_id>}

A(LLama3-8B): Failed
A(Mistral-7B): { "prediction": [ {"place_id": 1239, "probability": 0.12},..... ], "reason": "The prediction is based on the user's activity pattern and recent activities. Places with higher frequency in historical stays as well as context stays are given higher priority. Additionally, the analysis took into account the temporal information of the target stay, considering people's activity variations during different times and days." }
A(LLama3-70B): { "prediction": [12, 34, 56, 78, 90, 23, 45, 67, 89, 10], "reason": "Based on the user's activity pattern, recent activity, and time of day, the predicted next location is one of the places with ID 12, 34, 56, 78, 90, 23, 45, 67, 89, or 10." }
A(Mixtral-8x22B): Failed

A(DeepSeekV2): { "prediction": [1610, 3203, 850, 933, 1239, 1316, 202, 231, 2617, 287], "reason": "The user has a strong pattern of visiting place ID 1610 frequently, especially on Wednesdays and during the evening. The recent context shows visits to place ID 1610 on Monday evening, reinforcing this pattern. Additionally, place ID 3203 is also frequently visited, particularly during afternoon hours. The target stay's time (04:35 PM on Wednesday) aligns with the user's typical activity pattern, making these places highly probable." }
A(GPT3.5-Turbo): Failed
A(GPT4-Turbo): { "prediction": [1, 2, 3, 4, 5, 6, 7, 8, 9, 10], "reason": "Based on the user's history, they have frequent activities on Wednesday afternoons, particularly before dinner. Considering the recent context stays, it appears the user tends to visit specific places like cafes or malls on weekdays. Therefore, these locations are likely to be predicted." }

Figure 15: Prompt example for mobility prediction.

Boundary
The following is a multiple-choice question about the geospatial knowledge of city. Please choose the most suitable one among A, B, C and D as the answer to this question. Please output the option directly. No need for explaination.
Question: Which AOI is connected to Thistle Kensington Gardens by Bayswater Road?
A.Thistle Kensington Gardens
B.Portland Gallery
C.Cultural Section (Embassy of France)
D.Design Studio
Answer:

A(LLama3-8B): A
A(Mistral-7B): A. Thistle Kensington Gardens
A(LLama3-70B): C
A(Mixtral-8x22B): B.Portland Gallery

A(DeepSeekV2): C
A(GPT3.5-Turbo): A
A(GPT4-Turbo): A. Thistle Kensington Gardens

Figure 16: Prompt example for boundary related GeoQA.

District
The following is a multiple-choice question about the geospatial knowledge of city. Please choose the most suitable one among A, B, C and D as the answer to this question. Please output the option directly. No need for explaination.
Question: What is the address of Kensington Gardens?
A.on the east side of Haymarket, 100m from the southeast corner of the junction of Jermyn Street and Haymarket
B.on the east side of Aldersgate Street, 50m from the south corner of the junction of Aldersgate Rotunda and Aldersgate Street"
C.on the east side of Aldersgate Street, 50m from the south corner of the junction of Aldersgate Rotunda and Aldersgate Street
D.on the east side of Haymarket, within 50m from the east corner of the junction of Jermyn Street and Haymarket
Answer:

A(LLama3-8B): D
A(Mistral-7B): C. on the east side of Aldersgate Street, 50m from the south corner of the junction of Aldersgate Rotunda and Aldersgate Street (This option…)
A(LLama3-70B): D
A(Mixtral-8x22B): C. on the east side of Aldersgate Street, 50m from the south corner of the junction of Aldersgate Rotunda and Aldersgate Street

A(DeepSeekV2): A
A(GPT3.5-Turbo): D
A(GPT4-Turbo): A

Figure 17: Prompt example for districts related GeoQA.

**Landmark**
The fllowing is a multiple-choice question about the geospatial knowledge of city. Please choose the most suitable one among A, B, C and D as the answer to this question. Please output the option directly. No need for explaination.
Question: Which area of interest (AOI) is most likely to appear in the described environment among the following multiple AOIs?
A.Viet Food,Yaldham House
B.The Maple Leaf
C.The Maple Leaf
D.Moscos Cafe
E.Cafe,Salloos
Answer:

**A(LLama3-8B):** C. The Maple Leaf

**A(Mistral-7B):** D. Moscos Cafe. Without more context about the environment, a definitive answer cannot be given.
**A(LLama3-70B):** C
**A(Mixtral-8x22B): Failed**

**A(DeepSeekV2):** B.The Maple Leaf
**A(GPT3.5-Turbo):** B
**A(GPT4-Turbo):** B

Figure 18: Prompt example for landmark related GeoQA.

**Node**
**Q:** The following is a multiple-choice question about the geospatial knowledge of city. Please choose the most suitable one among A, B, C and D as the answer to this question. Please output the option directly. No need for explaination.
Question: What is the address of A Bar?
A. on the south side of Williamsburg Bridge, 1600m from the east corner of the junction of unknown road and Williamsburg Bridge
B. on the south side of Williamsburg Bridge, 1600m from the east corner of the junction of unknown road and Williamsburg Bridge
C. on the north side of Kent Avenue, 550m from the north corner of the junction of Broadway and Kent Avenue
D. on the west side of Kent Avenue, 450m from the north corner of the junction of Broadway and Kent Avenue
Answer:

**A(LLama3-8B):** C
**A(Mistral-7B):** C. on the north side of Kent Avenue, 550m from the north corner of the junction of Broadway and Kent Avenue
**A(LLama3-70B): Failed**
**A(Mixtral-8x22B): Failed**

**A(DeepSeekV2):** C
**A(GPT3.5-Turbo):** C.
**A(GPT4-Turbo):** C

Figure 19: Prompt example for node related GeoQA.

**Other**
The following is a multiple-choice question about the geospatial knowledge of city. Please choose the most suitable one among A, B, C and D as the answer to this question. Please output the option directly. No need for explaination.
Question: What is the land use type of Kensington Gardens?
A.Park&GreenLand
B.Entertainment,Residential
C.Residential
D.TrafficStation&Park
Answer:

**A(LLama3-8B):** A
**A(Mistral-7B):** A.Park&GreenLand
**A(LLama3-70B):** A
**A(Mixtral-8x22B):** A

**A(DeepSeekV2):** A
**A(GPT3.5-Turbo):** A
**A(GPT4-Turbo):** A

Figure 20: Prompt example for other GeoQA.

