# OpenReview forum: "CityBench: Evaluating the Capabilities of Large Language Models for Urban Tasks"
_ICLR.cc/2025/Conference — Submitted to ICLR 2025_

### Official Review · Reviewer_aRpA · 2024-10-18

**Soundness:** 4
**Presentation:** 4
**Contribution:** 4
**Rating:** 10
**Confidence:** 4

**Summary:**

This paper introduces **CityBench**, an innovative platform designed to evaluate the capabilities of large language models (LLMs) and visual language models (VLMs) in solving diverse urban tasks. CityBench integrates three core components: **CityData**, which collects and processes geospatial, visual, and human activity data; **CitySim**, a simulator that models urban dynamics like individual mobility, visual environments, and microscopic traffic behavior; and **CityBench**, which offers a comprehensive evaluation of eight urban tasks in two categories — perception & understanding and planning & decision-making.

The authors conducted extensive experiments involving 30 LLMs and VLMs across 13 global cities, evaluating tasks such as geospatial prediction, image geolocalization, traffic signal control, and mobility prediction. The results highlight the potential of LLMs and VLMs in handling urban tasks that require commonsense reasoning and semantic understanding, while also exposing challenges in tasks requiring precise professional knowledge and high-level reasoning, such as geospatial prediction and traffic control.

The paper contributes significantly by providing a scalable and systematic evaluation framework, uncovering strengths and limitations of current LLMs/VLMs in urban settings, and paving the way for further development and application of these models in real-world urban scenarios.

**Strengths:**

1. **High Level of Innovation**:
    - The paper introduces **CityBench**, a highly innovative and comprehensive evaluation platform. CityBench stands out by integrating multi-modal data and interactive simulations to systematically evaluate the performance of LLMs and VLMs in urban tasks, filling a significant gap in the evaluation benchmarks for urban research.
    - The combination of **CityData** and **CitySim** offers researchers a robust tool that simulates fine-grained dynamics across 13 cities. This large-scale, multi-city, multi-modal simulation environment sets CityBench apart from previous work, which has not addressed this level of complexity and scale.

2. **Comprehensive and Detailed Experiment Design**:
    - The experiment design is meticulous, covering both perception tasks (such as image geolocalization and infrastructure inference) and decision-making tasks (such as traffic signal control and mobility prediction). This creates a thorough evaluation framework that includes both straightforward perception tasks and more complex decision-making scenarios.
    - The authors conducted extensive experiments with 30 different LLMs and VLMs, providing a rich dataset and thorough result analysis, validating CityBench as an effective benchmark for urban research.
    - Additionally, the **error analysis and geospatial bias analysis** demonstrate a deep understanding of model behavior across different environments, offering valuable insights for future improvements.

3. **Practicality and Forward-looking Approach**:
    - The paper goes beyond theory by offering an open-source evaluation platform with easy-to-use APIs, significantly increasing its usability and scalability for further urban research applications.
    - It explores the potential of LLMs in urban research, particularly in dynamic and interactive environments, showcasing the future applications of LLMs in urban planning, traffic management, and other real-world urban scenarios.

**Weaknesses:**

1. **Limited Task Diversity**:
    - While the paper has designed a rich set of urban tasks, there is room for further expansion to include tasks that cover broader social, economic, and environmental aspects of urban research. Future iterations could consider incorporating more tasks related to community planning or disaster response, which would enhance CityBench’s applicability across a wider range of urban research domains.

**Questions:**

1. **Task Diversity**: One of the paper’s key strengths is the wide range of urban tasks included in CityBench. However, do the authors have plans to extend CityBench to cover tasks related to more social or environmental aspects of urban life, such as community planning or disaster response? Expanding the diversity of tasks could further enhance the platform’s applicability across various urban domains.

2. **Geospatial Bias**: The paper highlights the geospatial bias present in VLMs when evaluated across different cities. Could the authors provide further insight into why this bias exists? Specifically, what are the potential underlying factors (e.g., differences in urban morphology, data availability) contributing to this bias, and are there plans to mitigate it in future iterations of CityBench?

3. **Model Performance Variability**: The variability in performance across different LLMs and VLMs was an interesting finding, particularly the fact that larger models did not always outperform smaller ones. Could the authors elaborate on potential causes for this? Do the authors believe this is due to task complexity, model architecture, or data-specific issues?

4. **Error Handling**: The error analysis was quite revealing, especially the frequency of formatting and logic errors in certain models. Are there any specific strategies the authors would suggest to mitigate these types of errors in future deployments of LLMs and VLMs within urban tasks?

---

> ### Author Response · Authors · 2024-11-25
> **For Questions From Reviewer aRpA**
>
> **We sincerely thank the reviewer for their feedback and high evaluation of our work.**
>
> **For Q1：Task Diversity**
>
> Thank you for your valuable suggestions. We fully agree that diversifying tasks to incorporate more social, economic, and environmental dimensions in urban research is crucial. With access to the necessary data, we plan to enhance existing tasks by introducing sub-tasks such as carbon emission prediction and air quality forecasting within the geospatial prediction task, as well as landmark recognition within the ImageGeolocalization task. Furthermore, we aim to introduce new major tasks, such as Urban Spatiotemporal Perception, which will encompass urban event prediction and hotspot area detection. These additions are designed to provide broader coverage of social, economic, and environmental aspects, fostering a more comprehensive approach to urban research.
>
> **For Q2: Geospatial Bias**
>
> Thank you for your questions. Taking image geolocalization results as example, we provide simple evidence for this phenomenon by analyzing the number of public websites available on Google and Wikipedi in Sec A.3 in the appendix. Taking three well-performing cities (New York, London, Paris) and three under-performing cities (Shanghai, CapeTown, Nairobi) as examples, table below illustrates the relationship between the performance of street view image localization tasks and the size of the training corpus for LLMs. The training corpus size is approximated by the number of Google search entries and Wikipedia entries for each city. Compared to the well-performing cities, the under-performing cities have significantly smaller `training corpora' in the public websites. Additionally, open-source models exhibit significantly worse performance than commercial models in terms of geospatial bias. This observation provides initial evidence for analyzing geographical bias, but we believe there are more diverse factors contributing to this phenomenon. In the future, we plan to conduct larger-scale experiments on a more diverse set of cities to address this question and enhance performance based on the findings.
>
> | **Cities**   | **GPT4o** | **MiniCPM-V2.5** | **Google Search Pages** | **Wikipedia Pages** |
> | :--- | ---: | -----: | -----: | ----: |
> | **Shanghai** | 59%       | 0.4%       | 582,000,000       | 62,495        |
> | **CapeTown** | 81%       | 0     | 587,000,000       | 54,724        |
> | **Nairobi**  | 58.19%   | 0     | 205,000,000       | 14,226        |
> | **New York** | 94.39%   | 80%         | 7,070,000,000     | 1,039,264     |
> | **London**   | 88.80%   | 43.8%      | 5,460,000,000     | 848,381       |
> | **Paris**    | 92.80%   | 49%         | 4,170,000,000     | 353,500       |
>
> **For Q3: Model Performance Variability**
>
> Thank you for your suggestions. The performance variability across different LLMs and VLMs is primarily influenced by the capabilities of the underlying LLMs. For example, in widely used VLMs for urban tasks, Intern2.5-7B consistently outperforms Qwen2-7B and Mistral-7B in most tasks, which can be attributed to Intern2.5-7B's superior performance in general NLP tasks at the same parameter scale. For VLMs, larger models within the same architecture generally exhibit better performance. For instance, the performance typically follows the order: InternVL2-40B ≥ InternVL2-20B ≥ InternVL2-4B ≥ InternVL2-2B across most tasks. Similarly, LLaVA-NeXT-8B demonstrates performance comparable to MiniCPM-V2.5-8B, as both models leverage the same LLM backbone, LLaMA3-8B. Therefore, the observed variability in performance across different LLMs and VLMs is primarily attributed to the inherent capabilities of the underlying models. Thank you again for your thoughtful feedback!
>
> **For Q4: Error Handling**
>
> Thank you for your suggestions. Based on our experience, we recommend the following strategies to improve the robustness of models: 1. Enable robust data extraction: Utilize specific tools, such as json_repair, to enhance the reliability of information extracted from LLM outputs. 2. Task-specific training: Train models with task-specific data to minimize format errors and improve accuracy. 3. Enhanced instruction-following techniques: Adopt recent advancements in instruction-following methods[1,2] to develop more robust models. 4. Employ RAG (Retrieval-Augmented Generation) methods: Leverage RAG techniques to reduce hallucinations in LLMs, particularly for domain-specific knowledge. These approaches collectively aim to strengthen the robustness and reliability of models in diverse applications.
>
> [1] Dong, Guanting, et al. "Self-play with Execution Feedback: Improving Instruction-following Capabilities of Large Language Models." arXiv preprint arXiv:2406.13542 (2024).
>
> [2] Wu, Tianhao, et al. "Thinking LLMs: General Instruction Following with Thought Generation." arXiv preprint arXiv:2410.10630 (2024).

---

> ### Author Response · Authors · 2024-11-30
>
> **Dear Reviewer aRpA,**
>
> **We would like to ask if our response has addressed the concerns you raised. If further clarification or explanation is needed, we are more than happy to engage in further discussion. Once again, thank you for your invaluable feedback and high evaluation of our work!**

---

### Official Review · Reviewer_zAZV · 2024-10-26

**Soundness:** 2
**Presentation:** 2
**Contribution:** 1
**Rating:** 5
**Confidence:** 4

**Summary:**

The submission presents a framework to evaluate the capabilities of a series of foundation models, including some text-based LLMs and some multi-modal, say, image-text, models, in city-related tasks. It starts from data collection, data processing, and fianally carrying out tasks and benchmarking. It is no doubt the utility of LLM (in the broad sense including multi-modal) in analyzing human activities and city tasks is interesting. This could potentially shape the future research in LLM at large and the development of city-related LLM. However, the paper have not done a good job in providing concrete enough evidence in the underlying causes and lessons learnt. In addition, some experimental designs are flawed.

**Strengths:**

a. the utility of LLM for city-related tasks is an interesting point to explore
b. the benchmarking was done in a somewhat broad dataset with cities in different continents and a number of models are tested
c. the categorization of tasks is interesting and useful

**Weaknesses:**

a. the underlying causes of the good or bad performance of the models in different settings in terms of cities, models, tasks, are not discussed, only with vague and shallow discussions. This is supposed to be the key point to benefit future research and development. It could be ok that we do not push too much on novelty for this submission (more or less should be a major point for iclr though). However, the shallow explanations of the model performance and their comparison are disappointing. For instance, what is the root cause(s) of the unpleasant performance of LLM in regression tasks?

Indeed, the authors mentioned that LLM some cities do not have as good performance as other cities. The cause here is only “lesser-known”, without presenting any evidence of the underrepresentation. Other aspects, eg socio-economical, morphological, linguistic aspect, are not discussed.

The benchmarking paper can be benefitted a lot from deeper analysis and discussion. This is supposed to be the key.
b. LLM is in many ways not as good as, or only on par with conventional end-to-end or task specific methods. This somehow is interesting for developments in urban computing. But what is the lesson learnt here is rather sparsely mentioned.
c. the citydata module is only the introduction of datasets used, and it should not be listed as as a contribution brought up by this paper.

**Questions:**

a. the foursquare data is from 2016. Why not use more current data? Will the old dataset affect the findings?
b. it is unclear why the five qualities (the paper then interpreted the theory as six qualities) in “the image of the city” are used for GeoQA. The five qualities are proposed for guiding urban design practices. Could you elaborate on the relation between the five qualities and benchmarking GeoQA?

---

> ### Author Response · Authors · 2024-11-25
> **For Questions From Reviewer zAZV**
>
> **For Q1: the foursquare data is from 2016. Why not use more current data? Will the old dataset affect the findings?**
>
> Thank you for your valuable feedback. We apologize for the unclear description of the data selection process. Below, we provide a more detailed explanation of how the data was selected.
>
> **First**, the Foursquare dataset [1] stands as **one of the most influential global human mobility datasets of the past decade [2].** Although other datasets like Gowalla&Brightkite [3], Weeplace [4], and Yelp [5] also capture global mobility patterns, they predate Foursquare and offer more limited spatial coverage and data volume. Due to growing privacy considerations, the publication of mobility data has diminished, and most available trajectory datasets are restricted to individual cities or countries. For instance, the recent released YJMob100K dataset [6] (2024) is a synthetic data and encompasses only two Japanese cities. To address the need for broader geographical coverage and authentic mobility patterns, we utilize the widely used Foursquare dataset [1] for our global mobility analysis.
>
> [1] Yang, Dingqi, Daqing Zhang, and Bingqing Qu. "Participatory cultural mapping based on collective behavior data in location-based social networks." ACM TIST 2016.
>
> [2] Chen, Wei, et al. "Deep learning for trajectory data management and mining: A survey and beyond." arXiv preprint arXiv:2403.14151 (2024).
>
> [3] Cho, Eunjoon, Seth A. Myers, and Jure Leskovec. "Friendship and mobility: user movement in location-based social networks." KDD 2011.
>
> [4] Baral, Ramesh, and Tao Li. "Maps: A multi aspect personalized poi recommender system." Proceedings of the 10th ACM conference on recommender systems. 2016.
>
> [5] http://www.yelp.com/dataset_challenge
>
> [6] Yabe, Takahiro, et al. "YJMob100K: City-scale and longitudinal dataset of anonymized human mobility trajectories." Scientific Data 11.1 (2024): 397.
>
> **Second**, although the Foursquare dataset is somewhat outdated, both we and the research community [7-10] believe its impact on the findings of our benchmark is limited and acceptable. Following contemporary LLM-based mobility prediction studies [7,8], we convert locations into place IDs for our analysis, as shown in Figure 12 of the appendix. This abstraction enables the model to capture spatial-temporal patterns independent of specific location names, thus mitigating concerns about evolving spatial environments. Furthermore, **the dataset's continued adoption in recent mobility prediction research [7-10] demonstrates its enduring value for method evaluation.**
>
> [7] Wang, Xinglei, et al. "Where would i go next? large language models as human mobility predictors." arXiv preprint arXiv:2308.15197 (2023).
>
> [8] Feng, Shanshan, et al. "Where to move next: Zero-shot generalization of llms for next poi recommendation." 2024 IEEE Conference on Artificial Intelligence (CAI). IEEE, 2024.
>
> [9]Yang, Dingqi, et al. "Lbsn2vec++: Heterogeneous hypergraph embedding for location-based social networks." IEEE TKDE 2020
>
> [10] Feng, Jie, et al. "AgentMove: Predicting Human Mobility Anywhere Using Large Language Model based Agentic Framework." arXiv preprint arXiv:2408.13986 (2024).
>
> **For Q2: it is unclear why the five qualities (the paper then interpreted the theory as six qualities) in “the image of the city” are used for GeoQA. The five qualities are proposed for guiding urban design practices. Could you elaborate on the relation between the five qualities and benchmarking GeoQA?**
>
> Thank you for your insightful questions. While the concept was originally proposed to guide urban design practices, it also provides a systematic framework for understanding the key elements in urban space and has been widely used in spatially related research [1-3]. In CityBench, we adopt this approach to assess the spatial knowledge and capabilities of LLMs. Specifically, we use the concept of "The Image of the City" to categorize geospatial elements in urban spaces into distinct groups, enabling more detailed evaluation and analysis. The results of this categorization are presented in Table 6 in the appendix.
>
> [1] Pappalardo, Luca, et al. "Future directions in human mobility science." Nature computational science 3.7 (2023): 588-600.
>
> [2] Coutrot, Antoine, et al. "Entropy of city street networks linked to future spatial navigation ability." Nature 604.7904 (2022): 104-110.
>
> [3] Arribas-Bel, Daniel, and Martin Fleischmann. "Spatial Signatures-Understanding (urban) spaces through form and function." Habitat International 128 (2022): 102641.

---

> ### Author Response · Authors · 2024-11-25
> **For Weakness 1 From Reviewer zAZV**
>
> **For W1: the underlying causes of the good or bad performance of the models in different settings in terms of cities, models, tasks, are not discussed, only with vague and shallow discussions. ......**
>
> Thank you for your valuable suggestions. While conducting the experiments, we aimed to provide a more detailed analysis of the benchmark results. However, this has proven challenging due to the black-box nature of LLMs’ training corpora and weights. For instance, [1] provides a detailed analysis in a whole paper by only focusing on simple coordinate estimation tasks, but the diversity and complexity of urban tasks make such efforts significantly more difficult. As a result, we are currently only able to provide a high-level yet meaningful analysis. In the revised paper, we have added a more detailed analysis in **Section 4.2**.
>
> Here, we present a more detailed examination of the results, organized into four parts as follows.
>
> 1. Performance of VLMs on Urban Visual Taks. We present **additional results exploring the potential relationships between VLMs, LLMs, and their performance**. Our findings indicate that the performance variability across different LLMs and VLMs is primarily influenced by the capabilities of the LLM backbone. For instance, among widely used VLMs for urban tasks, Intern2.5-7B consistently outperforms Qwen2-7B and Mistral-7B in most tasks. This can be attributed to Intern2.5-7B's superior performance in general NLP tasks at the same parameter scale. Similarly, LLaVA-NeXT-8B demonstrates performance comparable to MiniCPM-V2.5-8B, as both models leverage the same LLM backbone, LLaMA3-8B.
> 2. Performance of LLMs on Urban Tasks without Visual Input. We present **additional results on the performance of VLMs on text-based tasks in Table 3**. Compared to their original LLM backbones, the performance of VLMs shows a noticeable decline. For example, CogVLM2-19B performs worse than Llama3-8B in both mobility prediction and urban exploration tasks. This highlights the importance of maintaining the general capabilities of LLMs during the training of VLMs to ensure their effectiveness across a wide range of tasks.
> 3. Error Analysis of VLMs. We present the **preliminary results of error analysis for VLMs in Figure 7**. Common errors identified include format errors, invalid actions, refusal to answer, and hallucinations. In the image geolocalization task, InternVL2-2B provides a response that do not follow the required format, while Yi-VL-34B gives an irrelevant invalid response. CogVLM2-19B and Yi-VL-34B, in the geospatial prediction and infrastructure inference tasks respectively, repeat the examples provided in the question and refuse to answer the actual question. Due to the response format requirements of the tasks, the most common error made by VLMs in urban visual tasks is misformatted responses.
> 4. Geospatial Bias of LLMs and VLMs. Taking image geolocalization results as example, **we provide simple evidence for this phenomenon by analyzing the number of public websites available on Google and Wikipedi in Sec A.3 in the appendix**. Taking three well-performing cities (New York, London, Paris) and three under-performing cities (Shanghai, CapeTown, Nairobi) as examples, table below illustrates the relationship between the performance of street view image localization tasks and the size of the training corpus for LLMs. The training corpus size is approximated by the number of Google search entries and Wikipedia entries for each city. Compared to the well-performing cities, the under-performing cities have significantly smaller `training corpora' in the public websites. Additionally, open-source models exhibit significantly worse performance than commercial models in terms of geospatial bias. This observation provides initial evidence for analyzing geographical bias, but we believe there are more diverse factors contributing to this phenomenon.
>
> | **Cities**   | **GPT4o** | **MiniCPM-V2.5** | **Google Search Pages** | **Wikipedia Pages** |
> | :--- | ---: | -----: | -----: | ----: |
> | **Shanghai** | 59%       | 0.4%       | 582,000,000       | 62,495        |
> | **CapeTown** | 81%       | 0     | 587,000,000       | 54,724        |
> | **Nairobi**  | 58.19%   | 0     | 205,000,000       | 14,226        |
> | **New York** | 94.39%   | 80%         | 7,070,000,000     | 1,039,264     |
> | **London**   | 88.80%   | 43.8%      | 5,460,000,000     | 848,381       |
> | **Paris**    | 92.80%   | 49%         | 4,170,000,000     | 353,500       |
>
> Finally, we are continuing our efforts to identify the root causes of LLM and VLM performance across these diverse urban tasks, models and cities. We will consider providing a more detailed and thorough analysis of them in future work.
>
> [1] Gurnee, Wes, and Max Tegmark. "Language Models Represent Space and Time." ICLR 2024.

---

> ### Author Response · Authors · 2024-11-30
> **For Weakness 2 and 3 From Reviewer zAZV**
>
> **For W2: LLM is in many ways not as good as, or only on par with conventional end-to-end or task specific methods. This somehow is interesting for developments in urban computing. But what is the lesson learnt here is rather sparsely mentioned**
>
> We appreciate your insightful feedback. While our paper provided limited discussion of this phenomenon due to the space limitation, our key evaluation findings align with recent vision papers [1,5], as outlined in our abstract and introduction. Below, we offer additional discussions:
> 1. Strengths of LLMs and VLMs in urban computing: These models perform well in certain scenarios, such as: 1) Tasks that require minimal numerical computation and where coarse geospatial information suffices, e.g., mobility prediction and inferring the city from a street view image. 2)Tasks demanding a high level of common sense and reasoning, e.g., infrastructure inference and outdoor navigation. For these tasks, developing methods oriented around LLMs and VLMs is promising, given their powerful foundational capabilities.
> 2. Challenges for LLMs and VLMs in urban computing: They struggle in scenarios requiring precise numerical capabilities, such as population density prediction and traffic signal control. In these tasks, LLMs and VLMs can be more effectively utilized by providing external semantic information about the original data as additional features, complementing traditional methods to achieve better results. For example, they can be fine-tuned to better understand and adapt to the patterns in these tasks, leading to improved results [1-4].
>
> [1] Mai, Gengchen, et al. "On the opportunities and challenges of foundation models for geoai (vision paper)." ACM Transactions on
> Spatial Algorithms and Systems (2024).
>
> [2] Letian Gong, et al. "Mobility-LLM: Learning Visiting Intentions and Travel Preference from Human Mobility Data with Large Language Models." NeurIPS 2024.
>
> [3] Li, Zhonghang, et al. "UrbanGPT: Spatio-temporal large language models." KDD 2024.
>
> [4] Manvi, Rohin, et al. "GeoLLM: Extracting Geospatial Knowledge from Large Language Models." ICLR2024
>
> [5] Zhang, Weijia, et al. "Urban foundation models: A survey." KDD 2024.
>
>
> **For W3: the citydata module is only the introduction of datasets used, and it should not be listed as as a contribution brought up by this paper.**
>
> Thank you for your suggestions. We apologize for the unclear description in the text. In our paper, we highlight CityData as one of our contributions for the following reasons:
> - First of all, we provide a variety of crawler implementations within CityData to facilitate easy and efficient data collection, enabling the extension of the benchmark to other cities.
> - More importantly, we offer preprocessing tools for these datasets, such as a map-building tool. This tool enhances open-source data to support subsequent simulations, including lane topology recovery, relationship recognition, intersection reconstruction, AOI mapping, POI clustering, basic traffic rule generation, and right-of-way construction. We have provided a more detailed description of this map-building tool in Figure 8 of the appendix.
>
> Thus, we present it alongside CitySim as one of the key contributions of our paper.

---

> ### Author Response · Authors · 2024-11-30
>
> **Dear Reviewer zAZV,**
>
> **We would like to ask if our response has addressed the concerns you raised. If further clarification or explanation is needed, we are more than happy to engage in further discussion. Once again, thank you for your hard work and invaluable feedback.**

---

> ### Comment · Area_Chair_ZmGd · 2024-12-01
>
> Dear Reviewer,
>
> Could you please look at the rebuttal by authors and see if your comments have been addressed?
>
> Thanks,
> AC

---

> ### Comment · Reviewer_zAZV · 2024-12-01
> **Response to authors’ rebuttal**
>
> Many thanks to the authors dedication to providing detailed responses to my questions and concerns. I reckon the analysis has been somewhat enhanced with the points in the rebuttal.
>
> I acknowledge that the authors have probed the difficulty of finding the underlying causes of the performance differences “due to the black-box nature of LLMs’ training corpora and weights”. And the authors have certain additions to the lessons learnt from the tests.
>
> That said, I continue to ponder the very core contribution of this submission to the communities of llm and perhaps urban computing. Without a shadow of doubt, it is good to know that some performing good while some not that good in these tasks and datasets. Whereas this type of benchmarking work is supposed to lay the groundwork for future developments. This contribution is then hardly visible with numerous figures and metrics presented. I'm still not sure what's the foundation laid from the submission.
> This resonates with the authors’ rebuttal “we are currently only able to provide a high-level yet meaningful analysis”. Apologies that I still like to push this a bit farther – the most meaningful messages for future development should be made clearer.
>
> A contribution that I definitely acknowledge is the framework being used in this paper. It is more systematic than others. At the end of the day, this becomes a question of whether a systematic test of llms, while accompanied by limited analysis and insights  due to the black-box nature of LLMs is good enough. I will leave this question to AC.
>
> I’ve been enjoying this conversation, and I’d certainly be happy to engage further.

---

> > ### Author Response · Authors · 2024-12-01
> >
> > Thank you very much for your insightful feedback and thoughtful suggestions. We sincerely appreciate your efforts in encouraging us to further deepen the analysis in our work.
> >
> > As you have noted, our systematic framework, along with its diverse tasks and broad datasets across cities, represents a significant contribution to the community. Additionally, we have made efforts to address most of your concerns, except for the challenging W1. While we acknowledge the concerns raised in W1, we have made every effort within our capabilities to refine and present the analysis in a way that addresses and reduces these concerns. Given this foundation, we feel that a score of 3 (rejection) may not fully capture the contributions of our paper to the community, nor does it seem to align with your partial recognition of our work. We would greatly appreciate it if you could reconsider the score and offer a more balanced evaluation of our paper.
> >
> > Lastly, we have made every effort to provide a detailed analysis presented above thus far and continue to work diligently to uncover the root causes of LLM and VLM performance variations across diverse urban tasks, models, and cities. Although we are somewhat disappointed with the current score, we sincerely apologize for not fully meeting your expectations regarding the depth of analysis. We genuinely appreciate your valuable suggestions and the effort you've put into helping us improve our work. Thank you once again for your constructive feedback and thoughtful engagement. We will continue to work towards developing a more comprehensive and in-depth benchmark and analysis to further contribute to the community.

---

> > > ### Comment · Reviewer_zAZV · 2024-12-02
> > >
> > > Could you please clearly point out what is the groundwork laid here for future development of llm and urban computing? Appreciated.

---

> ### Author Response · Authors · 2024-12-02
>
> Thank you for giving us the opportunity to highlight the contributions of our work to the future development of LLMs and urban computing. Below, we summarize our contributions to these two research communities:
>
> 1. From the perspective of LLMs:
> - While robust evaluation of LLMs is a critical aspect of their development, we present an easy-to-use, out-of-domain and dynamic benchmark designed to help researchers identify potential general issues in current models, such as hallucinations and limitations in instruction-following capabilities, thereby contributing to the improvement of LLM generalization for real-world scenarios.
> - Furthermore, considering the bias issues in LLMs, our work provides a comprehensive dataset for evaluating the geospatial biases of LLMs, contributing to their fairness and enhancing their utility across diverse cultural contexts.
> 2. From the perspective of Urban Computing:
> - We present the first systematic evaluation results of LLMs across a range of urban computing tasks, offering valuable insights into their limitation and potential within this domain.
> - Furthermore, our benchmark serves as an easy-to-use and powerful platform to enable researchers in urban computing to develop domain-specific LLMs capable of effectively addressing urban challenges.
>
> In summary, our benchmark serves as a pivotal step in advancing research efforts and driving innovation in leveraging LLMs to develop cutting-edge intelligence for urban computing. Thank you once again for your constructive feedback and thoughtful insights. We remain committed to advancing our work by developing a more comprehensive and in-depth benchmark and analysis to further contribute to the community.

---

> ### Comment · Reviewer_zAZV · 2024-12-02
>
> In fact, what I wanted to see is the takeaway of the work to future developments of llms.
>
> That said, I will raise my score to acknowledge the authors' dedication.

---

> > ### Author Response · Authors · 2024-12-03
> >
> > Thank you for your thoughtful feedback and for raising the score. We are pleased that our revisions addressed some of your concerns. Moving forward, we are dedicated to advancing our work by developing a more comprehensive and in-depth benchmark and analysis, aiming to provide meaningful takeaways for the future development of LLMs and contribute further to the Urban Computing community.

---

### Official Review · Reviewer_UmiM · 2024-10-28

**Soundness:** 3
**Presentation:** 2
**Contribution:** 3
**Rating:** 6
**Confidence:** 4

**Summary:**

This paper targets the limitations in current explorations of large language models (LLMs) within the urban domain. It creatively introduces an interactive simulator to construct a multi-task, multi-dimensional benchmark and conducts extensive, comprehensive experiments, yielding intriguing conclusions. The authors release their code to foster the development of the community.

**Strengths:**

1. It’s exciting to see the promising potential of methods based on interactive simulators in the field of LLMs for urban applications; this work can provide a meaningful boost to community development.

2. The workload is substantial, covering a wide range of tasks with detailed and thorough experiments.

**Weaknesses:**

1. The paper provides a somewhat vague description of the pipeline. For example, in Section 3.3.1, what are the criteria for identifying low-quality data? What are the standards for filtering and rewriting? If data quality remains unsatisfactory after filtering and rewriting, is it filtered again or further processed in the same manner?

    In Section 3.1 CityData, the authors state that OSM data is unsuitable for city data construction and introduce a globally applicable rule-based map construction tool. However, there is no specific description of this tool, which leaves readers uncertain about its details and limits understanding of its functionality.

2. A suggestion: the name 'CitySim' is already in use[1], so choosing a different name might help reduce potential ambiguity.

3. In the construction of the Human Activities Data in Section 3.1, data from the literature published in 2016 was used. Can data from a paper published eight years ago accurately reflect the current state of human activities in cities?

4. There is another work CityEval in [2] published in June 2024, which also evaluates the capabilities of LLMs in urban spaces, and some tasks are similar to those in this work. Has the author considered comparing and discussing the differences between the two?

[1] Zheng, Ou, et al. "CitySim: a drone-based vehicle trajectory dataset for safety-oriented research and digital twins." Transportation research record 2678.4 (2024): 606-621.

[2] Feng, Jie, et al. "CityGPT: Empowering Urban Spatial Cognition of Large Language Models." arXiv preprint arXiv:2406.13948 (2024).

**Questions:**

Please see weakness.

---

> ### Author Response · Authors · 2024-11-25
> **For Weakness 1 and 2 From Reviewer UmiM**
>
> **For W1.1: The paper provides a somewhat vague description of the pipeline.For example, in Section 3.3.1, what are the criteria for identifying low-quality data? What are the standards for filtering and rewriting? If data quality remains unsatisfactory after filtering and rewriting, is it filtered again or further processed in the same manner?**
>
> Thank you very much for your thoughtful suggestions. In CityBench, the authors will participate in the quality control process for some tasks, following the automatic quality control stage. Taking the manual checking of GeoQA as an example, the original data from OpenStreetMap contains low-quality information, with missing or incorrect details about AOI, POI, and roads. When this low-quality data is used in the evaluation task, LLMs may become confused and generate meaningless answers. In such cases, the authors review the questions to ensure that the information in the context is meaningful. However, due to time limitations for participants, we can only randomly sample the evaluation cases. For instances from cities and regions where authors are unfamiliar, we filter out low-quality instances. For instances from cities and regions where authors are familiar, we rewrite low-quality instances using external information, such as commercial map services. Finally, if data quality remains unsatisfactory after filtering and rewriting, we will regenerate a certain number of cases to fill the gaps. In fact, while considering the probabilities of filtering, we generate enough candidate instances during the initial generation.
>
> **For W1.2: In Section 3.1 CityData, the authors state that OSM data is unsuitable for city data construction and introduce a globally applicable rule-based map construction tool. However, there is no specific description of this tool, which leaves readers uncertain about its details and limits understanding of its functionality.**
>
> Thank you very much for your thoughtful suggestions. We have added **more description and Figure 8** about the map-building tool in the appendix of revised paper. Here, we introduce the basic information of map-building tool as below. This tool enhances open-source map data to support subsequent behavior simulations, encompassing lane topology recovery, relationship recognition, intersection reconstruction, area of interest (AOI) mapping, point of interest (POI) clustering, basic traffic rule generation, and right-of-way construction. You can refer **Figure 8** in the appendix for details.
>
> **For W2: A suggestion: the name 'CitySim' is already in use[1], so choosing a different name might help reduce potential ambiguity.**
>
> Thank you for your careful suggestion. Following your advice, we have renamed CitySim in our paper to CitySimu to distinguish it from the published paper in 2024.

---

> ### Author Response · Authors · 2024-11-25
> **For Weakness 3 and 4 From Reviewer UmiM**
>
> **For W3: ......, data from the literature published in 2016 was used. Can data from a paper published eight years ago accurately reflect the current state of human activities in cities?**
>
> Thanks for your valuable feedback. We choose this data due to the following reasons.
>
> **First**, the Foursquare dataset [1] stands as **one of the most influential global human mobility datasets of the past decade [2].** Although other datasets like Gowalla&Brightkite [3], Weeplace [4], and Yelp [5] also capture global mobility patterns, they predate Foursquare and offer more limited spatial coverage and data volume. Due to growing privacy considerations, the publication of mobility data has diminished, and most available trajectory datasets are restricted to individual cities or countries. For instance, the recent released YJMob100K dataset [6] (2024) is a synthetic data and encompasses only two Japanese cities. To address the need for broader geographical coverage and authentic mobility patterns, we utilize the widely used Foursquare dataset [1].
>
> [1] Yang, Dingqi, Daqing Zhang, and Bingqing Qu. "Participatory cultural mapping based on collective behavior data in location-based social networks." ACM TIST 2016.
>
> [2] Chen, Wei, et al. "Deep learning for trajectory data management and mining: A survey and beyond." arXiv:2403.14151 (2024).
>
> [3] Cho, Eunjoon, Seth A. Myers, and Jure Leskovec. "Friendship and mobility: user movement in location-based social networks." KDD 2011.
>
> [4] Baral, Ramesh, and Tao Li. "Maps: A multi aspect personalized poi recommender system." RecSys. 2016.
>
> [5] http://www.yelp.com/dataset_challenge
>
> [6] Yabe, Takahiro, et al. "YJMob100K: City-scale and longitudinal dataset of anonymized human mobility trajectories." Scientific Data 11.1 (2024): 397.
>
> **Second**, although the Foursquare dataset is somewhat outdated, both we and the research community [7-10] believe its impact on the findings of our benchmark is limited and acceptable. Following contemporary LLM-based mobility prediction studies [7,8], we convert locations into place IDs for our analysis, as shown in Figure 12 of the appendix. This abstraction enables the model to capture spatial-temporal patterns independent of specific location names, thus mitigating concerns about evolving spatial environments. Furthermore, **the dataset's continued adoption in recent mobility prediction research [7-10] demonstrates its enduring value for method evaluation.**
>
> [7] Wang, Xinglei, et al. "Where would i go next? large language models as human mobility predictors." arXiv preprint arXiv:2308.15197 (2023).
>
> [8] Feng, Shanshan, et al. "Where to move next: Zero-shot generalization of llms for next poi recommendation." 2024 IEEE Conference on Artificial Intelligence (CAI). IEEE, 2024.
>
> [9]Yang, Dingqi, et al. "Lbsn2vec++: Heterogeneous hypergraph embedding for location-based social networks." IEEE TKDE 2020
>
> [10] Feng, Jie, et al. "AgentMove: Predicting Human Mobility Anywhere Using Large Language Model based Agentic Framework." arXiv preprint arXiv:2408.13986 (2024).
>
>
>
> **For W4: There is another work CityEval in [2] published in June 2024........ Has the author considered comparing and discussing the differences between the two?**
>
> Thank you for your valuable suggestions. We have cited this paper and revised the relevant sections in the paper. We provide a detailed discussion of them below. Compared to CityEval, which focuses solely on text-based evaluations, our benchmark is designed for **multi-modal evaluation** of LLMs and VLMs. Thus, **the four evaluation tasks with visual input in our benchmark are entirely different from those in CityEval.** Additionally, our benchmark is distributed across **13 cities worldwide**, whereas most tasks in CityEval cover only three cities, and some tasks are limited to a single city(e.g., only reporting results in Beijing for three composite tasks). Regarding the four text-based tasks, the **traffic signal task is unique** to our benchmark. The other three tasks—GeoQA, mobility prediction, and urban exploration—may appear similar to those CI, mobility prediction and spatial navigation in CityEval, but they differ in two key aspects: 1) our benchmark covers **13 cities worldwide and includes results from 30+ LLMs and VLMs.; 2) the datasets and methods used in these tasks are different**, taking the mobility prediction task as an example, we use global data from Foursquare-checkin[1] and follow the methodology of LLM-Mob [2] for predictions. In contrast, CityEval does not specify its data source and is limited to testing in Beijing without prompt details.
>
> [1] Yang, Dingqi, Daqing Zhang, and Bingqing Qu. "Participatory cultural mapping based on collective behavior data in location-based social networks." ACM TIST 2016.
>
> [2] Wang, Xinglei, et al. "Where would i go next? large language models as human mobility predictors." arXiv preprint arXiv:2308.15197 (2023).

---

> ### Author Response · Authors · 2024-11-30
>
> **Dear Reviewer UmiM,**
>
> **We would like to ask if our response has addressed the concerns you raised. If further clarification or explanation is needed, we are more than happy to engage in further discussion. Once again, thank you for your hard work and invaluable feedback.**

---

### Official Review · Reviewer_ApVj · 2024-11-03

**Soundness:** 3
**Presentation:** 3
**Contribution:** 3
**Rating:** 6
**Confidence:** 5

**Summary:**

This paper introduces CityBench, a systematic benchmark for evaluating the capabilities of large language models (LLMs) in urban research. The authors develop CityData and CitySim to integrate diverse urban data and simulate urban dynamics, constructing CityBench with eight representative tasks. The findings indicate that while advanced LLMs excel in understanding human dynamics and semantic inference in urban images, they struggle with complex tasks requiring specialized knowledge and high-level reasoning, such as geospatial prediction and traffic control. These insights offer valuable perspectives for the future application and development of LLMs.

**Strengths:**

1. Open Source: The authors have developed CityBench and CitySim, which involve a substantial amount of engineering effort, and have also made the codebase open-source. This is beneficial for future research in this domain.
2. Rich geographic diversity: CityBench covers 13 cities, providing a rich geographic diversity.

**Weaknesses:**

1. Figure presentation issues: The left panel of Figure 5 aims to showcase differences in performance across cities, but lacks a legend to identify different cities.
2. Experiment: 1) Incomplete and lacking insight in Error Analysis: The error analysis section only provides insights into LLM errors, with no analysis of VLM errors. Additionally, the LLM error analysis merely highlights general issues common across benchmarks, such as instruction-following limitations and hallucinations, without offering task-specific insights relevant to CityBench’s urban-focused tasks. A more robust analysis could involve systematically sampling each model’s errors on individual tasks, with detailed analysis and conclusions. 2) Incomplete test results: Both Table 2 and Table 3 present CityBench results. While Table 2 omits LLM results due to the visual input requirement, Table 3 involves purely language-based tasks, making it feasible to test VLMs as well. Including VLM results for these tasks would provide a more comprehensive evaluation.
3. Method：1) Transparency of CityBench data volume: Table 1 provides extensive data details for CityData; however, the main text does not include specific data quantities for CityBench. Additionally, the appendix lacks clear information on the exact number of tasks, providing only approximate figures. 2) Unclear task setup: The distinction between testing LLMs and VLMs separately for the Outdoor Navigation and Urban Exploration tasks is not well-explained. According to the descriptions, these tasks are similar, both involving decision-making. However, only LLMs are tested in one task and VLMs in the other, without a clear rationale.

**Questions:**

1、Vague description of data processing: The quality control section mentions the role of human annotators and states that the authors participated in data filtering and rewriting. However, the standards and proportions for filtering and rewriting are not specified. Including these details would make CityBench more transparent and easier to understand.

---

> ### Author Response · Authors · 2024-11-25
> **For Questions and Weakness1 From Reviewer ApVj**
>
> **For Q1: Vague description of data processing: The quality control section mentions the role of human annotators ……**
>
> Thank you very much for your thoughtful suggestions. In CityBench, the authors will participate in the quality control process for some tasks, following the automatic quality control stage. Taking the manual checking of GeoQA as an example, the original data from OpenStreetMap contains low-quality information, with missing or incorrect details about AOI, POI, and roads. When this low-quality data is used in the evaluation task, LLMs may become confused and generate meaningless answers. In such cases, the authors review the questions to ensure that the information in the context is meaningful. However, due to time limitations for participants, we can only randomly sample 10% of the evaluation cases. For instances from cities where authors are unfamiliar, we filter out low-quality instances. For instances from cities where authors are familiar, we rewrite low-quality instances using external information, such as assistance with commercial map services. We have updated these decsriptions in the appendix of the paper to helper reader better understand our work.
>
> **For W1: Figure presentation issues: The left panel of Figure 5 aims to showcase differences in performance across cities, but lacks a legend to identify different cities.**
>
> Thank you for your careful suggestions, and we apologize for the lack of clarity in the legend of Figure 5. In the original paper, the order of the cities in the left part of Figure 5 was noted within its caption. To enhance clarity, following your suggestions, we have now added a new legend to the left panel of Figure 5. You can refer to the updated Figure 5 in the paper. Additionally, we provide the city order here for your reference: Tokyo, Paris, London, Mumbai, Moscow, Cape Town, Nairobi, San Francisco, São Paulo, Shanghai, Sydney, New York, and Beijing.

---

> ### Author Response · Authors · 2024-11-25
> **For Weakness3 From Reviewer ApVj**
>
> **For W3.1: Method, Transparency of CityBench data volume**
>
> Thank you for your careful suggestions. Based on your feedback, we have updated the **Table 6** in the revised paper to include the exact number of instances in the entire benchmark. The revised table is also presented below.
>
> | CityBench     | Tasks  | Modality | Metrics     | Instances   | Images |
> | :------- | :-----: | :------: | :-----: | :----: | :----: |
> | Perception&Understanding  | Image Geolocalization    | Image    | Acc, Acc@1km/25km       | 6500  | 6500   |
> |   | Geospatial Prediction    | Image    | r 2 , RMSE  | 5739  | 5739   |
> |   | Infrastructure Inference | Image    | Accuracy, Recall  | 5739  | 5739   |
> |   | GeoQA for City Elements  | Text     | Accuracy    | 13126       | /      |
> | Planning& Decision Making | Mobility Prediction      | Text     | Top1-Acc, F1      | 6500  | /      |
> |   | Urban Exploration  | Text     | Steps, Success Rate     | 650   | /      |
> |   | Outdoor Navigation       | Image    | Distance, Steps, Success Rate | 650   | 55984  |
> |   | Traffic Signal Control   | Text     | Queue Length, Throughput      | 1hour × 13 cities | /      |
>
> **For W3.2: Method, Unclear task setup**
>
> For the tasks Urban Exploration and Outdoor Navigation, both types of tasks indeed involve decision-making, but the biggest difference between them lies in modality. The former only involves a single modality of text, while the latter incorporates the modality of street view images. This is why we only test VLMs for the Outdoor Navigation task. However, it is truly not very appropriate to test only LLMs on the Urban Exploration task. We have supplemented the results of VLMs on all text-only tasks (including Urban Exploration). Due to time constraints, at this stage we have only added the results of some VLMs, but we assure that the next version of the paper will include the results of all relevant VLMs. The detail information of tasks Urban Exploration and Outdoor Navigation are shown below:
>
> ```python
> ### Urban Exploration:
> Your navigation destination is Square André Lefèvre. You are now on Quai des Grands Augustins, nearby POIs are: Fontaine Wallace. Given the available options of road and its corresponding direction to follow and corresponding direction, directly choose the option that will help lead you to the destination. The options are: A ('Boulevard Saint-Michel', 'Northeast') B ('Quai Saint-Michel', 'East') C ('Boulevard Saint-Michel', 'East'). Directly make a choice.
>
> ### Outdoor Navigation:
> Navigate to the described target location!
> Action Space: forward, left, right, stop
> - If you choose "forward", proceed for 50 meters.
> - If you choose "left" or "right", make the turn at the next intersection.
> - If you believe you have reached the destination, please select "stop".
> - Please respond ONLY with "forward", "left", "right", or "stop".
> Navigation Instructions: xxx
> Previous action: xxx
> Here is the street view image of your current location.<image>
> Please provide the next action('forward', 'left', 'right', or 'stop') based on the image and the navigation instruction:
> ```

---

> ### Author Response · Authors · 2024-11-30
> **For Weakness2 From Reviewer ApVj**
>
> **For W2.1: Experiment, Incomplete and lacking insight in Error Analysis**
>
> Thank you for your valuable suggestions. We have included analysis of reprentative errors in VLMs for vision-based tasks in additional **Figure 7** in the revised paper. In the original version, we identified the most common issues in the error analysis of benchmark results. However, as urban tasks are translated into a language format, we are still working to establish task-specific error definitions that do not rely solely on language-based error descriptions.
>
> **For W2.2: Experiment, Incomplete test results**
>
> Thank you for your valuable suggestions. After several days of dedicated effort, we have provided additional results of VLMs on text-based tasks, which are now included in **Table 3** of the revised paper. For your convenience, we also present these results below. Compared to their original LLM backbones, VLMs exhibit a noticeable decline in performance. For example, CogVLM2-19B underperforms Llama3-8B in both mobility prediction and urban exploration tasks. This highlights the importance of preserving the general capabilities of LLMs during the training of VLMs, making it a crucial direction for ensuring their effectiveness across diverse tasks.
>
> |  | **Tasks**  | **Understanding** |  | | **Planning Decision-making** |  |    |  |
> | :----- | :------- | :-----: | :------: | :---: | :------: | :----: | :-----: | :---: |
> | |  | **GeoQA**  | **Mobility Prediction**  | | **Urban Exploration**  |   | **Traffic Signal**  |  |
> | | **Metrics**  | **Accuracy** | **Top1 Acc** | **F1** | **Succ Rate** | **Steps**| **Queue** | **Throughput** |
> | | **LSTPM**   | -   | 0.114 | 0.086  | -     | -   | -   | -      |
> | **Baselines** | **Fixed-Time**      | -    | - | -   | -     | -   | 57.870  | 993.333    |
> |   | **Max-Pressure**    | -    | - | -   | -     | -   | **36.898**  | **1345.333**   |
> |   | **Mistral-7B**      | 0.229    | 0.090   | 0.087  | 0.730     | 5.382   | 64.120  | 853.333    |
> |   | **Qwen2-7B** | 0.289    | 0.142   | 0.109  | 0.697     | 5.889   | 62.271  | 880.000    |
> |   | **Intern2.5-7B**   | 0.304    | 0.118   | 0.102  | 0.738     | 5.552   | 55.121  | 1047.667   |
> |   | **LLama3-8B**       | 0.297    | 0.130   | 0.094  | 0.747     | 5.304   | 57.738  | 1014.333   |
> |   | **Gemma2-9B**       | 0.339    | 0.131   | 0.120  | 0.716     | 5.679   | 74.475  | 651.333    |
> | **LLMs**      | **Intern2.5-20B**       | 0.315    | 0.116   | 0.098  | 0.679     | 6.243   | 61.229  | 958.667    |
> |   | **Gemma2-27B**      | 0.349    | 0.145   | 0.118  | 0.713     | 5.733   | 56.081  | 1009.333   |
> |   | **Qwen2-72B**       | 0.357    | 0.155   | 0.135  | 0.697     | 5.887   | 66.924  | 793.333    |
> |   | **LLama3-70B**      | 0.329    | **0.159**    | 0.130  | **0.796**     | **4.941**   | 59.338  | 959.667    |
> |   | **Mixtral-8x22B**   | 0.321    | 0.155   | **0.136**     | 0.745     | 5.339   | 65.682  | 821.333    |
> |   | **DeepSeekV2**      | 0.358    | 0.126   | 0.101  | 0.698     | 5.739   | 56.086  | 1020.333   |
> |   | **InternVL2-2B**     | 0.296    | 0.000    | 0.000     | 0.672     | 6.015   | 55.725  | 1012.000   |
> |   | **InternVL2-4B**     | 0.304    | 0.130    | 0.102     | 0.674     | 6.091   | 74.499  | 647.667    |
> |   | **InternVL2-8B**     | 0.329    | 0.142    | 0.102     | 0.703     | 5.714   | 53.196  | 1069.667   |
> |   | **InternVL2-26B**    | 0.310    | 0.137    | 0.107     | 0.694     | 5.723   | 57.512  | 971.667    |
> |   | **InternVL2-40B**    | 0.351    | 0.159    | 0.121     | 0.675     | 6.041   | 52.459  | 1087.000   |
> | **VLMs**      | **Qwen2VL-2B**       | 0.293    | 0.103    | 0.075     | 0.643     | 6.315   | 56.097  | 1003.667   |
> |   | **Qwen2VL-7B**       | 0.286    | 0.144    | 0.102     | 0.660     | 6.155   | 55.885  | 995.333    |
> |   | **MiniCPM-V2.5-8B** | 0.308    | 0.124    | 0.092     | 0.708     | 5.643   | 56.066  | 1001.000   |
> |   | **LLaVA-NeXT-8B**    | 0.313    | 0.124    | 0.084     | 0.688     | 5.891   | 56.184  | 989.333    |
> |   | **GLM-4v-9B**   | 0.296    | 0.133    | 0.092     | 0.680     | 5.979   | 53.870  | 1058.000   |
> |   | **CogVLM2-19B**      | 0.282    | 0.026    | 0.029     | 0.710     | 5.905   | 55.229  | 1046.667   |
> |   | **GPT3.5-Turbo**   | 0.285    | 0.152   | 0.113  | 0.719     | 5.473   | 56.219  | 1022.000   |
> |   | **GPT4-Turbo**      | **0.398**    | 0.147   | 0.125  | 0.757     | 5.184   | 55.761  | 1022.333   |

---

> ### Author Response · Authors · 2024-11-30
>
> **Dear Reviewer ApVj,**
>
> **We would like to ask if our response has addressed the concerns you raised. If further clarification or explanation is needed, we are more than happy to engage in further discussion. Once again, thank you for your hard work and invaluable feedback.**

---

> ### Comment · Area_Chair_ZmGd · 2024-12-01
>
> Dear Reviewer,
>
> Could you please look at if the authors have addressed your concerns?
>
> Thanks,
> AC

---

> > ### Comment · Reviewer_ApVj · 2024-12-01
> >
> > The authors have addressed most of my concerns, I raised my score to 6.

---

> > > ### Author Response · Authors · 2024-12-01
> > >
> > > Thank you for your thoughtful review and for raising the score. We are grateful that our revisions have addressed most of your concerns. Your constructive feedback has been invaluable in helping us improve the quality of our work.

---

### Official Review · Reviewer_sDWi · 2024-11-04

**Soundness:** 3
**Presentation:** 4
**Contribution:** 3
**Rating:** 8
**Confidence:** 3

**Summary:**

The paper releases a new systematic benchmark to evaluate LLM and VLM capabilities on geospatial urban data, including interactive and non-interactive tasks that test semantic understanding as well as reasoning.

**Strengths:**

**(S1)**: A comprehensive benchmark with multiple interactive and non-interactive tasks. The authors provide a benchmark that has a good diversity of tasks to evaluate VLM and LLM capabilities. The tasks that form this benchmark are valuable to the research community and for future LLM developers.

**(S2)**: Extensive evaluations with popularly used large language models. The experiments break down the performance for each task for multiple LLMs with sensible metrics. (although I think Llama 405B is missing).

**(S3)**: Publicly released code and API for running the evaluation. It is good that the authors already have the code and API ready for this benchmark, which should aid reproducibility and adoption for CityBench.

Overall, I think the benchmark provided in this paper is valuable to the research community for the diversity of tasks it includes. The paper is well-written and presented, and so I lean towards acceptance.

**Weaknesses:**

**(W1)**: Template based questions. The authors specify in section 3.3.1 that they use LLMs to generate instructions for tasks and also use LLMs to filter out low-quality data. More details are needed here, such as which LLMs are used, whether a mixture of LLMs are used, and how much this biases the downstream benchmark performance in favor of certain models. In general, more specific details on the quality control process are required (eg: how much low-quality data was generated, how much was manually curated etc.).

**(W2)**: Not enough geographic diversity. While the authors do make an effort to collect data for 13 cities across the globe, as they note in Figure 5, the performance on less renowned cities is worse for most LLMs. I think for a more comprehensive benchmark, the authors could consider including a larger diversity of cities for more fine-grained analysis on LLM capabilities.

**(W3)**: Missing comparisons with human baselines. I am curious to know how well humans perform on some of these interactive tasks and it would be good to include a baseline for human-level performance within the benchmark.

**Questions:**

**(Q1)**: Figure 5: Why is the performance on New York so low, considering it is a very well-known city. Are there any hypotheses for this case?

**(Q2)**: How exactly are satellite/street-view images sampled for the dataset? What is the distribution of these images per city in the benchmark?

**(Q3)**: On examples where the model always provides an answer (i.e. not ones where it refuses to answer), how do the different models compare?

**(Q4)**: Did you try evaluating few-shot performance for these models? How does LLM performance scale with in-context examples?

---

> ### Author Response · Authors · 2024-11-25
> **For Questions From Reviewer sDWi**
>
> **For Q1: Figure 5: Why is the performance on New York so low, considering it is a very well-known city. Are there any hypotheses for this case?**
>
> Thank you for your detailed feedback. Upon carefully reviewing the experimental results, we discovered an error in the plotting code where an incorrect version of the results was used, leading to incomplete data for several cities, including New York. After correcting this mistake, we have updated Figure 5 in the paper, showing that the performance for New York is comparable to other well-known cities. We sincerely appreciate your meticulous feedback once again. Additionally, we have conducted a thorough review of the other results to ensure no similar errors are present.
>
> **For Q2: How exactly are satellite/street-view images sampled for the dataset? What is the distribution of these images per city in the benchmark?**
>
> Thank you for your thoughtful questions. We acquired full-coverage satellite imagery at zoom level 15 for each city and applied a sampling strategy to optimize benchmark usage. For cities with fewer than 500 satellite images, all images were included; for those with more, 500 images were randomly sampled. Similarly, we generated 10,000 random coordinate points within each city and used the Google Street View API to obtain street view images, excluding points where imagery was unavailable. The same sampling strategy was applied: if fewer than 500 images were available, all were included; otherwise, 500 were randomly sampled. The spatial distribution of satellite and street view images for each city is detailed in the appendix.
>
> **For Q3: On examples where the model always provides an answer (i.e. not ones where it refuses to answer), how do the different models compare?**
>
> Thank you for your careful feedback. In our opinion, refusal cases are similar to misformatted cases and are treated as incorrect in the metrics calculations. Due to the diverse formats of refusals across different models and tasks, it is not feasible to directly identify and exclude them when calculating the performance of the remaining instances. As a result, we can only include a selective subset of these cases to illustrate the potential issues with underperforming models. We are continuing our efforts to identify these cases and will report the results excluding refusal cases. Thank you for your understanding.
>
> **For Q4: Did you try evaluating few-shot performance for these models? How does LLM performance scale with in-context examples?**
>
> Thank you for your valuable suggestions. Due to the limitation of time and computing power, we present the few-shot performance of several representative LLMs in Beijing in the following table. For all text-based tasks, we use 2-shot as the default few-shot method. However, we do not provide few-shot results for vision-based tasks, as many VLMs are unable to handle the multi-image requirements. As shown in the table, the impact of few-shot learning varies across different models and tasks. For instance, few-shot learning improves performance for Gemma-27B in GeoQA but reduces it for Gemma2-9B on the same task. Similarly, it benefits Gemma2 in traffic signal tasks but proves detrimental for Llama3. In the future, we plan to provide more comprehensive results on the few-shot performance across various models and tasks.
>
> | **model@Beijing**       | **GeoQA** | **mobility** |         | **urban exploration** |           | **traffic signal** |                 |                  |
> | :------: | :-------: | :------: | :-----: | :--------: | :-------: | :--------: | :-------: | :------: |
> |                         | **acc**   | **acc1**     | **f1**  | **success rate**      | **steps** | **Throughput**     | **Travel Time** | **Queue Length** |
> | **Gemma2-9B-fewshot**   | 0.3064   | 0.1146      | 0.1009 | 0.728                | 6.152    | 1528               | 2129.21        | 64.269          |
> | **Gemma2-9B-zeroshot**  | 0.339    | 0.131       | 0.120  | 0.716                | 5.679    | 1448               | 2214.451       | 71.729          |
> | **Gemma2-27B-fewshot**  | 0.3593   | 0.1085      | 0.0765 | 0.696                | 6.26     | 2240               | 1746.89        | 31.683          |
> | **Gemma2-27B-zeroshot** | 0.349    | 0.145       | 0.118  | 0.713                | 5.733    | 2187               | 1762.182       | 33.322          |
> | **LLama3-8B-fewshot**   | 0.2884   | 0.0954      | 0.0546 | 0.692                | 6.424    | 1547               | 2132.233       | 61.84           |
> | **LLama3-8B-zeroshot**  | 0.297    | 0.130       | 0.094  | 0.747                | 5.304    | 2128               | 1873.757       | 40.941          |
> | **Llama3-70B-fewshot**  | 0.3427   | 0.0886      | 0.0620 | 0.796                | 4.941    | 1810               | 1962.493       | 50.541          |
> | **Llama3-70B-zeroshot** | 0.329    | 0.159       | 0.130  | 0.74                 | 5.876    | 2031               | 1893.517       | 43.475          |

---

> ### Author Response · Authors · 2024-11-25
> **For Weaknesses From Reviewer sDWi**
>
> **For W1: Template based questions. The authors specify in section 3.3.1 that they use LLMs to generate instructions for tasks and also use LLMs to filter out low-quality data. More details are needed here, such as which LLMs are used, whether a mixture of LLMs are used, and how much this biases the downstream benchmark performance in favor of certain models. In general, more specific details on the quality control process are required (eg: how much low-quality data was generated, how much was manually curated etc.).**
>
> Thank you very much for your thoughtful suggestions. In CityBench, the authors will participate in the quality control process for some tasks, following the automatic quality control stage. Taking the manual checking of GeoQA as an example, the original data from OpenStreetMap contains low-quality information, with missing or incorrect details about AOI, POI, and roads. When this low-quality data is used in the evaluation task, LLMs may become confused and generate meaningless answers. In such cases, the authors review the questions to ensure that the information in the context is meaningful. However, due to time limitations for participants, we can only randomly sample the evaluation cases. For instances from cities and regions where authors are unfamiliar, we filter out low-quality instances. For instances from cities and regions where authors are familiar, we rewrite low-quality instances using external information, such as commercial map services. Finally, if data quality remains unsatisfactory after filtering and rewriting, we will regenerate a certain number of cases to fill the gaps. In fact, while considering the probabilities of filtering, we generate enough candidate instances during the initial generation.
>
> **For W2: Not enough geographic diversity. While the authors do make an effort to collect data for 13 cities across the globe, as they note in Figure 5, the performance on less renowned cities is worse for most LLMs. I think for a more comprehensive benchmark, the authors could consider including a larger diversity of cities for more fine-grained analysis on LLM capabilities.**
>
> Thank you for your valuable suggestions. We agree that including a greater diversity of cities (far more than 13, for example, 100) in the benchmark would be highly beneficial for more fine-grained analysis of LLM capabilities. Unfortunately, due to constraints on time and computational resources, we are unable to provide results from a larger number of cities at this time. In the future, we will strive to expand the benchmark to include more cities (e.g., 100 as a milestone) with greater geographic diversity, enhancing its comprehensiveness. By open-sourcing the entire pipeline, we also warmly invite the community to contribute to these results for promoting the development of the field.
>
> **For W3: Missing comparisons with human baselines. I am curious to know how well humans perform on some of these interactive tasks and it would be good to include a baseline for human-level performance within the benchmark.**
>
> Thank you for your insightful suggestions. Our benchmark consists of eight tasks encompassing over 20,000 instances, making it impractical for humans to label them directly due to the large volume. Given the constraints of time and cost, it is challenging to recruit enough experts to establish human baselines for these tasks. In the future, we plan to develop a human labeling tool to engage the community in collecting sufficient results to serve as human baselines.

---

> > ### Comment · Reviewer_sDWi · 2024-11-29
> > **Response to Author Rebuttal**
> >
> > Thank you for your rebuttal and for fixing the errors with NYC. I am curious to know the answer to Q3 when you have those results ready.
> >
> > For W1-- I appreciate the work that goes into quality control, but it would be valuable to have a few quantitative metrics on this process. Further, I am still missing answers to these questions: "More details are needed here, such as which LLMs are used, whether a mixture of LLMs are used, and how much this biases the downstream benchmark performance in favor of certain models". Quantitative metrics for questions like these "how much low-quality data was generated, how much was manually curated etc." would be valuable.
> >
> > For W2, I understand the time constraint, and it is ok to not have those 100 cities ready in the next few days. I will look forward to  the geographical expansion in the future!
> >
> > For W3-- again I agree it is difficult in time and resources to get human evaluation numbers. However, any kind of oracle or northstar for the benchmark would be valuable to know how much LLMs can reasonable improve their performance on CityBench. I'm curious to know what you think of this, if any such northstar exists or even if it makes sense.

---

> > > ### Author Response · Authors · 2024-12-01
> > > **For Follow-up Quesitions From Reviewer sDWi**
> > >
> > > Thank you very much for your kind suggestions and valuable feedback.
> > >
> > > **For Q3: On examples where the model always provides an answer (i.e. not ones where it refuses to answer), how do the different models compare?**
> > >
> > > After several days of dedicated effort, we have made progress in addressing your concerns. Given the large volume of tasks and models, we selected one visual task (Image Geolocalization) and one textual task (Traffic Signal Control) for analysis, focusing on several representative models. To begin, we manually identified refusal error cases and subsequently designed prompts to enable LLMs (Qwen2.5-72B) to detect these refusal errors automatically. While we have verified the consistency between human-identified and LLM-identified refusal errors, there is still the possibility that some potential refusal errors were overlooked. We will continue to refine our approach to ensure more detailed and robust results in future work. The findings of our current analysis are summarized as follows.
> > >
> > > We first present the error distribution for Image Geolocalization Task. As shown in Table 1, the refusal rate across all models is low. Consequently, the performance of the models, as reported in Table 2 without refusal, is comparable to their performance with refusal included.
> > >
> > > Table 1
> > >
> > > | **Model Name**   | **Normal** | **Refusal** | **Uninformed** | **Invalid Action** | **Misformatted** |
> > > | :- | :-: | :-: | :--: | :--: | :--: |
> > > | **InternVL2-2B** | 64.11%   | 0.21%  | 30.01%    | 0.10% | 5.56%   |
> > > | **InternVL2-40B**    | 94.82%   | 0.02%  | 0.00% | 0.00% | 5.16%   |
> > > | **Qwen2-VL-2B** | 77.87%   | 0.00%  | 22.13%    | 0.00% | 0.00%   |
> > > | **Yi\_VL\_34B**  | 7.97%    | 0.12%  | 90.89%    | 0.38% | 0.64%   |
> > >
> > > Table 2
> > > | **Model**    | **City Acc(with refusal)** | **City Acc(w/o refusal)** |
> > > | :- | :--: | :-: |
> > > | **Qwen2-VL-2B** | 0.630 | 0.630    |
> > > | **InternVL2-2B** | 0.238 | 0.239    |
> > > | **Yi\_VL\_34B**  | 0.251 | 0.252    |
> > > | **InternVL2-40B**    | 0.574 | 0.574    |
> > >
> > > For the Traffic Signal Control task, we provide the error distribution in the table below. Unlike the previous task, the refusal rate for these models is significantly higher, reflecting the complexity of the task. However, due to the nature of this task, where performance metrics are determined after multi-step processes requiring long-term observation windows, it is not feasible to directly report model performance after removing refusal errors.
> > >
> > > | **Model**  | **Normal** | **Refusal** | **Uninformed** | **Invalid Action** | **Misformatted** |
> > > | :-- | :-: | :-: | :--: | :--: | :--: |
> > > | **gemma-2-27b**    | 80.06%   | 6.90%  | 5.27% | 3.21% | 4.55%   |
> > > | **llama3-8b**  | 87.08%   | 0.42%  | 11.46%    | 1.04% | 0.00%   |
> > > | **Qwen2-72B**  | 92.89%   | 0.00%  | 7.05% | 0.06% | 0.00%   |
> > > | **Qwen-Qwen2-7B** | 81.70%   | 3.10%  | 14.38%    | 0.74% | 0.09%   |
> > >
> > >
> > > **For W1(follow-up): "More details are needed here, such as which LLMs are used, whether a mixture of LLMs are used, and how much this biases the downstream benchmark performance in favor of certain models" ......**
> > >
> > > We use GPT-4omini as the default LLM for quality assessment in the CityBench without incorporating a diverse mix of other LLMs. This reliance on GPT-4omini introduces potential biases, particularly the risk of overestimating its performance within the benchmark. Such biases reflect one of the inherent limitations of the LLM-as-judge approach in automated evaluations. Given the recent advancements and practices in LLM-as-judge research [1], we plan to revise the automated quality control process in the next version of CityBench. This revision will include a thorough analysis of quality control biases of different models and the implementation of quantitative metrics to rigorously address these issues.
> > >
> > > [1] Gu, Jiawei, et al. "A Survey on LLM-as-a-Judge." arXiv preprint arXiv:2411.15594 (2024).
> > >
> > >
> > > **For W3(follow-up): ....... However, any kind of oracle or northstar for the benchmark would be valuable to know how much LLMs can reasonable improve their performance on CityBench........**
> > >
> > > Thank you so much for your thoughtful and insightful question. We fully agree with your perspectives. Defining a northstar for CityBench would be invaluable for understanding the performance ceiling of large language models and guiding future research directions. We are genuinely interested in pursuing such a goal, while also acknowledging the significant challenges posed by the inherent diversity and complexity of the tasks involved. Building on our current work, we plan to take steps toward this objective by developing a more focused, hard-version subset of CityBench, leveraging human annotation and fostering collaboration within the broader community. Once again, we sincerely appreciate your valuable suggestion, which has provided us with a deeper appreciation of the importance of our work and clearer guidance for its future direction!

---

> > > > ### Comment · Reviewer_sDWi · 2024-12-02
> > > > **Feedback on Author Rebuttal**
> > > >
> > > > Most of my concerns are addressed and so I am inclined to increase my score. The score increase assumes that the authors will carry out their promised revisions to CityBench as stated in the above response, including:
> > > > * a complete analysis of task-specific refusal-rates and how that impacts the metrics and also indicates the difficulty of different tasks relevant to each other
> > > > * a more complete analysis of the bias introduced by using GPT-4o or other models to generate template-based questions/answers, including improvements to the automated quality control process and an analysis with quantitative metrics that the authors mentioned above.
> > > > * introduction of the CityBench-Hard subset which will help provide a northstar for performance on this benchmark.
> > > >
> > > > Thank you to the authors for their work on this paper.

---

> > > > > ### Author Response · Authors · 2024-12-03
> > > > >
> > > > > Thank you so much for your valuable suggestions and for raising the score. We sincerely appreciate that our revisions have addressed most of your concerns. Your constructive feedback has been instrumental in enhancing the quality of our work, and we are truly grateful for your thoughtful input. We are fully committed to further improving our work by implementing the plan outlined based on your suggestions.

---

### Meta-Review · Area_Chair_ZmGd · 2024-12-23

**Metareview:**

The paper proposes CityBench, a comprehensive suite to evaluate the capabilities of LLMs for urban tasks. While acknowledging the importance of LLMs and/or VLMs for understanding the urban tasks under a wide variety of adaptations, there is barely a systematic and scalable evaluation benchmark. The paper is more like a comprehensive evaluation / study report of LLMs into the urban tasks.

Overall this is a good and well-written paper. Most reviewers aknowledge the good merits of the paper:

- Comprehensive, large-scale, high-quality dataset. Promised to be open-sourced to the community.
- CityData and CitySimu, are two contributed modules that are designed to support multiple applications with easy-to-use APIs.

There are also some potential negative comments around the preliminary manuscript:

- Missing experiments for some results in Table 2, 3. Lack of insight.
- Important technical details to construct the data.

Authors did a fair enough job to address most concerns raised by reviewers. AC carefully read all the review comment, rebuttal and discussions, and thought the scope of the manuscript might not be appropriate for the time being. In a nutshell, this work is more curated to be an LLM evaluation report, rather than technical findings. There might be limitations on the insight / motivation behind all the observations.

There are several crucial concerns that need to be raised before further consideration.

(a) Whether to contribute to future research is still in doubt. AC agrees with the opinion that is raised by one of the reviewers that the future potential follow-up research might be limitted. Knowing well all the pros and cons of the LLMs, how to apply to urban tasks is still under-explored. Such a report might be a good advide to decision makers, how would it benefit from the real-applications is not convincing. That being said, AC is not judging this kind of research is not useful; but rather would argue there might be limitted impact for the ICLR or similar venue community.

(b) [most crucial point to affect the decision] how each LLM behave is uncertain, due to the black-box nature of LLMs. As such, there might be limited novelty (technical contribution) if there are few insights / rationale behind the behaviors of LLMs. This is actually raised by **all** reviewers in some extent. To name a few:

- Are there any specific strategies the authors would suggest to mitigate these types of errors in future deployments of LLMs and VLMs within urban tasks?

- what is the root cause(s) of the unpleasant performance of LLM in regression tasks?

- Can data from a paper published eight years ago accurately reflect the current state of human activities in cities?

To sum up, the paper is indeed a good investigation with comprehensive amount of work, which indeed is much appreciated. And yet might not be appriopriate for the audience at ICLR.

**Additional Comments On Reviewer Discussion:**

While four reviewers send out very good comments on the manuscript, AC steps in, read carefully on the paper and discussed with reviewers. Overall the paper might not draw great interest if only LLMs are evaluated for facts.  Note: AC do acknowledge the high rating by reviewers and have identified each review comment independantly.

---

### Decision · Program_Chairs · 2025-01-22

Reject